# Ultra-high-granularity detector simulation with intra-event aware generative adversarial network and self-supervised relational reasoning

Baran Hashemi[1] ✉, Nikolai Hartmann[2], Sahand Sharifzadeh[3], James Kahn[4,5] & Thomas Kuhr [●][2]

Simulating high-resolution detector responses is a computationally intensive process that has long been challenging in Particle Physics. Despite the ability of generative models to streamline it, full ultra-high-granularity detector simulation still proves to be difficult as it contains correlated and fine-grained information. To overcome these limitations, we propose Intra-Event Aware Generative Adversarial Network (IEA-GAN). IEA-GAN presents a Transformer-based Relational Reasoning Module that approximates an event in detector simulation, generating contextualized high-resolution full detector responses with a proper relational inductive bias. IEA-GAN also introduces a Self-Supervised intra-event aware loss and Uniformity loss, significantly enhancing sample fidelity and diversity. We demonstrate IEA-GAN's application in generating sensor-dependent images for the ultra-high-granularity Pixel Vertex Detector (PXD), with more than 7.5 M information channels at the Belle II Experiment. Applications of this work span from Foundation Models for high-granularity detector simulation, such as at the HL-LHC (High Luminosity LHC), to simulation-based inference and fine-grained density estimation.

The Efficient and Fast Simulation[1–8] campaign in particle physics sparked the search for faster and more storage-efficient simulation methods for collider physics experiments. Simulations play a vital role in various downstream tasks, including optimizing detector geometries, designing physics analyses, and searching for new phenomena beyond the Standard Model (SM). Efficient detector simulation has been revolutionized by the introduction of the Generative Adversarial Network (GAN)[9] for image data.

Deep Generative Models have been widely used in particle physics to achieve detector simulation for the LHC[2–4,8,10–15], mainly targeting calorimeter simulation or collision event generation[16–21]. Previous work on generating high spatial resolution detector responses includes the

Prior-Embedding GAN (PE-GAN) by Hashemi et al.[7], which utilizes an end-to-end embedding of global prior information about the detector sensors, and the work by Srebre et al.[6] in which a Wasserstein GAN[22] with gradient penalty[23] was used as a proof of concept to generate high-resolution images without conditioning. For mid-granularity calorimeter simulation, the recent approaches[8,12], experiment with several GAN-like and Flow-based[24] architectures with <30k simulated channels, and 3DGAN[11] for high granularity calorimeter simulation with only 65k pixel channels. Nonetheless, these studies barely scratch the surface of the profound challenges posed by future detector simulations. Take, for instance, the impending High Granularity Calorimeter (HGCAL) - a component of the High Luminosity Large Hadron Collider

[1]ORIGINS Data Science Lab, Technical University Munich, Munich, Germany. [2]Faculty of Physics, Ludwig Maximilians University in Munich, Munich, Germany. [3]Faculty of Computer Science, Ludwig Maximilians University in Munich, Munich, Germany. [4]Helmholtz AI, Karlsruhe, Germany. [5]Steinbuch Centre for Computing (SCC), Karlsruhe Institute of Technology (KIT), Karlsruhe, Germany. ✉e-mail: baran.hashemi@origins-cluster.de

(HL-LHC)[25] upgrade program at the Compact Muon Solenoid (CMS) experiment[26]. With an estimated 6.5 million detector channels distributed across 50 layers, the HGCAL's complexity far surpasses the capacity of existing methods, pointing to the urgency of developing more advanced simulation approaches.

The task of learning to generate ultra-high-resolution detector responses has several challenges. First, in general, we are dealing with spatially asymmetric high-frequency hitmaps. With current state-of-the-art (SOTA) GAN setups for high-resolution image generation candidates, when the discriminator becomes much stronger than the generator, the fake images are easily separated from real ones, thus reaching a point where gradients from the discriminator vanish. This happens more frequently with asymmetric high-resolution images due to the difficulty of generating imbalanced high-frequency details. On the other hand, a less powerful discriminator results in a mode collapse, where the generator greedily optimizes its loss function, producing only a few modes to deceive the discriminator.

Furthermore, the detector responses in an event, a single readout window after the collision of particles, share both statistical and semantic similarities[27] with each other. For example, the sparsity (occupancy) of each image within a class, defined as the fraction of pixels with a non-zero value, shows statistical similarities between detector components (see the Supplementary Figs in supplementary note). Moreover, as the detector response images show extreme resemblance at the semantic and visual levels[27], they can be classified as fine-grained images. When generating fine-grained images, the objective is to create visual objects from subordinate categories. A similar scenario in computer vision is generating images of different dog breeds or car models. The small inter-class and considerable intra-class variation inherent to fine-grained image analysis make it a challenging problem[28]. The current state-of-the-art conditional GAN models focus on class and intra-class level image similarity, in which intra-image[29], data-to-class[30], and data-to-data[31] relations are considered. However, in the case of detector simulation, classes become hierarchical and fine-grained, and the discrimination between generated classes that are semantically and visually similar becomes harder. Therefore, the aforementioned models show extensive class confusion[32,33] at the inter-class level. In addition, since the information in an event comes from a single readout window of the detector, the processes happening in this window affect all sensors simultaneously, leading to a correlation among them (see "Results" section). In this paper, we demonstrate how this fine-grain intra-event correlation plays a pivotal role in the downstream Physics analysis.

To overcome all these challenges with ultra-high-resolution detector simulation, we introduce the Intra-Event Aware GAN (IEA-GAN), a novel deep generative model to generate sensor/layer-dependent detector response images with the highest fidelity while satisfying all relevant metrics. Since we are dealing with a fine-grained and contextualized (by each event) set of images that share information. First, we introduce a Relational Reasoning Module (RRM) for the discriminator and generator to capture inter-class relations. Then, we propose a loss function for the generator to imitate the discriminator's knowledge of dyadic class-to-class correspondence[34]. Finally, we introduce an auxiliary loss function for the discriminator to leverage its reasoning codomain by imposing an information uniformity condition[35] to alleviate the mode-collapse issue and increase the generalization of the model. IEA-GAN captures not only statistical-level and semantic-level information but also a correlation among samples in a fine-grained image generation task.

We demonstrate the IEA-GAN's application on the ultra-high dimensional data of the Pixel Vertex Detector (PXD)[36] at Belle II[37] with more than 7.5M pixel channels- the highest spatial resolution detector simulation dataset ever analyzed with deep generative models. Then, we investigate several evaluation metrics and show that in all of them, IEA-GAN is in much better agreement with the target distribution than

other SOTA deep generative models for high-dimensional image generation. We also perform an ablation study and exploration of hyperparameters to provide insight into the model.

It is crucial to highlight that our approach extends beyond the scope of the existing models in calorimeter simulations that try to capture layer-by-layer correlations[38]. While these existing models do consider observables that depend on more than one layer simultaneously, they are restricted to simulating particle showers originating from a single particle source within a confined and localized area of the shower. In contrast, our approach embraces the complexity of an entire event with multiple-particle origins, encompassing the full detector simulation. This perspective offers to capture correlations between detector sensors across various angles and layers. By doing so, it approximates the intricate and dynamic interplay of sensors throughout the entire detector, surpassing the limitations of simulations focused solely on localized particle showers.

In this paper, we study the most challenging detector simulation problem with the highest spatial resolution dataset coming from the Pixel Vertex Detector (PXD)[36], shown in Fig. 1, the innermost tracking sub-detector of Belle II[37]. The configuration of the PXD consists of 40 sensors within two detector layers, as shown in Fig. 1. The inner layer has 16 sensors, and the outer layer comprises 24 sensors. Thus, each event includes 40 gray-scale images, each with a resolution of $250 \times 768$ pixels, resulting in more than 7.5 million pixel channels per event as shown in Fig. 1. The recorded background signatures by PXD that comprise the majority of the PXD hits in each event come from various processes in the detector that do not originate from the physics processes of interest, called signal processes. These background processes can be categorized into beam-induced and luminosity-dependent processes. The beam-induced processes come from the synchrotron radiation and collisions of beam particles with residual gas in the beampipe, bending magnets, or particles within a bunch. In contrast, luminosity-dependent processes comprise electron-positron collisions leading to physics processes such as Bhabha scattering or two-photon processes.

The problem with such a high-resolution background overlay[39] is that many resources are required for their readout, storage, and distribution. For example, the size of the PXD background overlay data needed for the simulation of a single event is ~200 kB. This is roughly $2N$ times the size of the background overlay data per event with respect to all other detector components together[40] where $N$ is the PXD background amplifier coefficient. Thus, an idea is to simulate them online. However, the on-the-fly simulation of background events is not feasible due to the considerable simulation time required by Geant4, which takes approximately 1500 seconds to simulate a single event. As a result, while storing such a massive amount of data is very inefficient for high-resolution detectors, we propose to generate the background signatures with IEA-GAN on the fly as a surrogate model.

Here we show, by applying IEA-GAN, to the PXD at Belle II, we are able not only to reduce the storage demand for pre-produced background data by a factor of 2 (see "Discussion" section) but also enable us to have the ability of online simulation as shown in Fig. 1 by dramatically reduce the CPU time of online simulation in comparison to the old infeasible Geant4 approach. As a result, It is now finally possible to employ the IEA-GAN as an online surrogate model for the ultra high-granularity PXD background simulation on the fly, a task that was unattainable before for such a high-resolution detector simulation. Thus, IEA-GAN stands as the viable candidate capable of managing the ultra-high granularity of the forthcoming HL-LHC[25] era.

## Results
### IEA-GAN architecture
A Generative Adversarial Network (GAN) is an unsupervised deep learning architecture that involves two networks, the Generator and the Discriminator, whose goal is to find a Nash equilibrium[41] in a two-

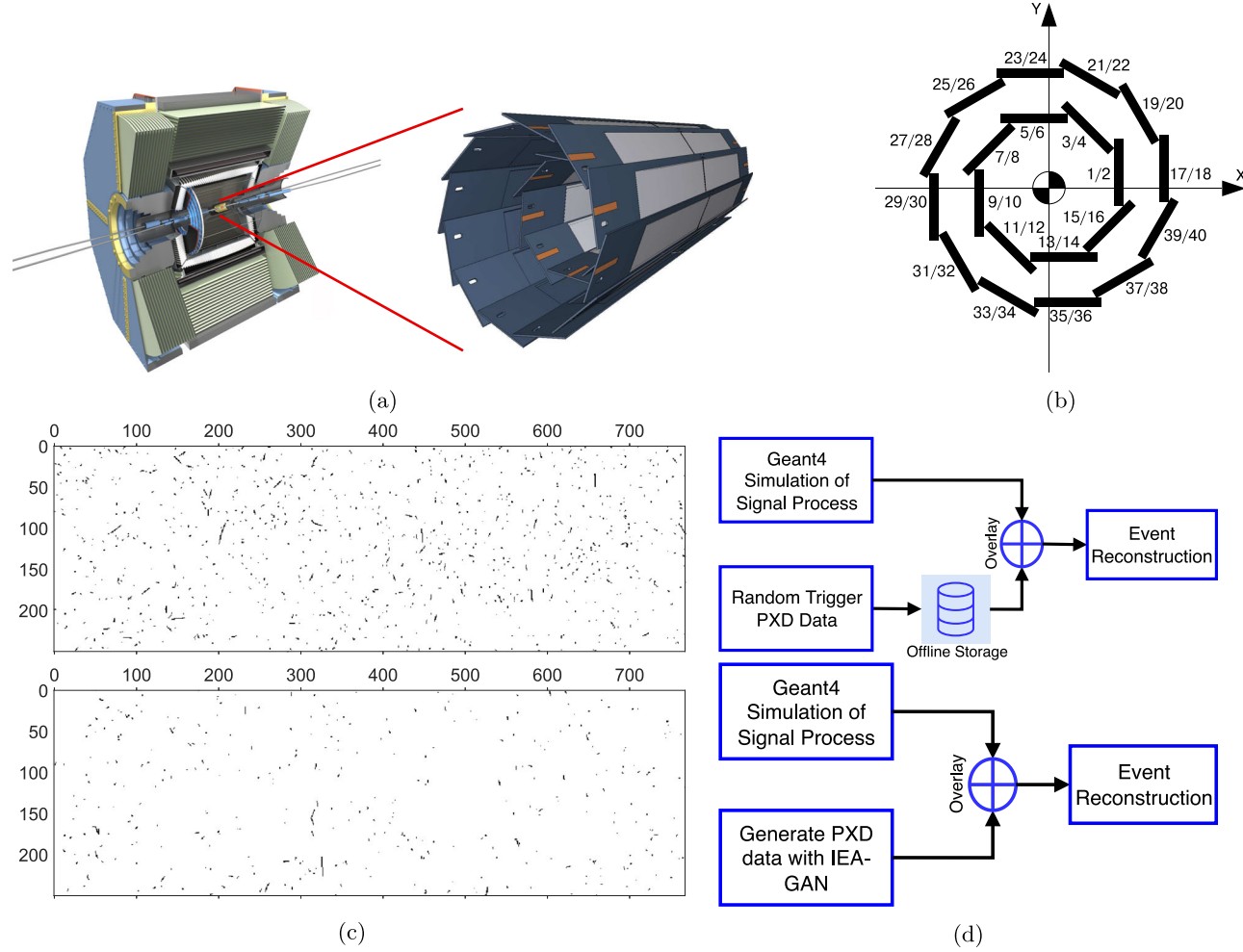

Fig. 1 | Pixel detector layout. (Taken from ref. 100). The pixel detector (PXD) is the inner-most sub-detector of the Belle II experiment (**a**) and is configured in a two-layered overlapping sensor structure (**b**). PXD image examples (**c**) for sensors 7 (top) and 25 (down). **d** The event generation pipeline with Geant4[52] (top) and using IEA-GAN (bottom). Generating PXD data on the fly of analysis avoids the need to store them offline.

player min-max problem. IEA-GAN, as shown in Fig. 2, is a deep generative model based on a self-supervised relational grounding.

IEA-GAN's discriminator, D, takes the set of detector response images $x_i \in \mathbf{R}^d$ coming from one event and embeds them as input nodes within a fully connected event graph in a self-supervised way. The Event graph is a weighted graph where the nodes are the embedded detector images in an event, and the edges are weighted by the degree of similarity between the detector images in each event (see "Methods" section). IEA-GAN approximates the concept of an event (the event graph) by contextual reasoning using the permutation equivariant Relational Reasoning Module (RRM). RRM is a GAN-compatible, fully connected, multi-head Graph Transformer Network[42–44] that groups the image tokens in an event based on their contextual similarity. The contextual degree of similarity between samples in an event is learned by the attention mechanism in the RRM. For multimodal contrastive reasoning, the discriminator also takes the sensor embedding of the detector as class tokens (see "Methods" section). In the end, it compactifies both image and class modalities information by projecting the normalized graph onto a hypersphere as discussed in detail in section 4.

To ensure that the Generator G has a proper understanding of an event and captures the intra-event correlation, it first samples from a Normal distribution, $\mathcal{N}(0,1)$, at each event as random degrees of freedom (Rdof), and decorates the sensor embeddings with this four-dimensional learnable Rdof (see "Methods" section). Then, for a self-

supervised contextual embedding of each event, the RRM acts on top of this. Notably, Rdof differs from the original GAN[9] Gaussian latent vectors. Rdof can be considered as an event-level learnable segment embedding[45] or perturbation[46] to the token embeddings, which can leverage the diversity of generated images. Combining these modules with the IEA Loss allows the Generator to gain insight and establish correlation among the samples in an event, thus improving its overall performance.

Apart from the adversarial loss, IEA-GAN also benefits from a self-supervised and contrastive-based set of losses. The model understands the geometry of the detector through a proxy-based contrastive 2C loss[31] where the learnable proxies are the sensor embeddings over the hypersphere. Moreover, to improve the diversity and stability of the training, we introduce a Uniformity loss for the discriminator. The Uniformity loss can encourage the discriminator to give equal weight to all regions of the hypersphere[35] rather than just focusing on the areas where it can easily distinguish between real and fake data. Encouraging the discriminator to impose uniformity not only promotes more diverse and varied outputs but also mitigates issues such as mode collapse.

Another essential part of IEA-GAN is the IEA loss that addresses the class confusion[32,33] problem of the conditional generative models for fine-grained datasets. In the IEA-loss, the generator tries to imitate the discriminator's understanding of each event through a dyadic information transfer with a stop-gradient (sg) for the discriminator. This can

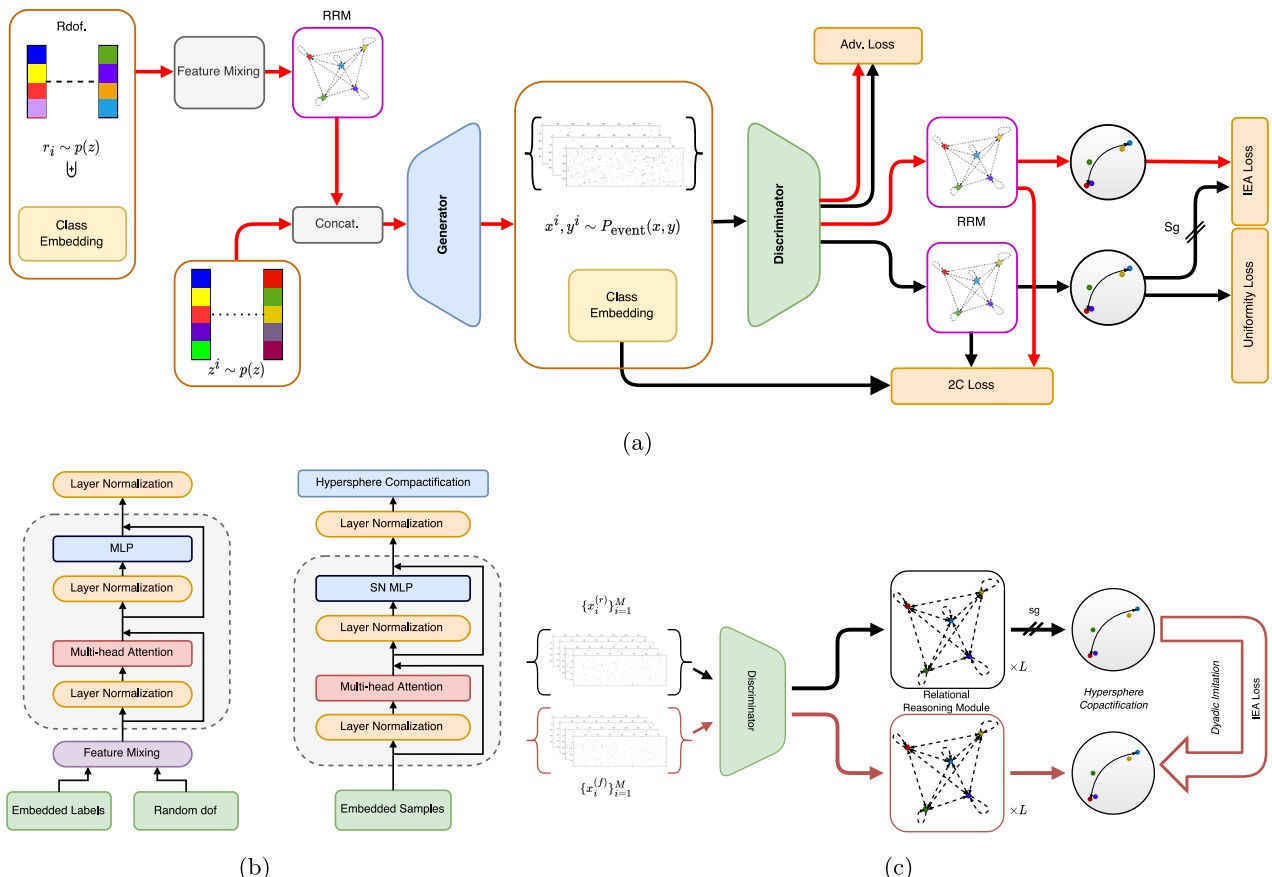

**Fig. 2 | Pixel detector layout, and the simulation pipelines.** IEA-GAN architecture (**a**) and Relational Reasoning Module components (**b**), and the IEA Self-Supervised Loss (**c**). **a** Rdof stands for Random degrees of freedom, which decorates the generator's sensor/layer embedding with an event-level learnable embedding responsible for the generator's intra-event correlation. The Relational Reasoning Modules (RRM) in the generator and the discriminator do the intra-event reasoning by clustering class/image embeddings based on their contextual similarity, respectively. The red lines correspond to the forward and backward passes of the generator. The black lines correspond to the forward and backward passes of the discriminator. The discriminator is trained with the Adversarial loss, see Eq. (6), 2C loss, see Eq. (7) and the Uniformity loss, see Eq. (17). On the other hand, the generator uses the Adversarial loss, 2C loss, and the IEA loss, see Eq. (16), illustrated in **c**. Sg means stop-gradient for the discriminator from the IEA loss, a self-supervised dyadic-aware loss for the generator. **b** The Relational Reasoning Module (RRM) for the generator (left) and the discriminator (right) create event graphs at each iteration. The attention mechanism inside the RRM learns the contextual degree of similarity between samples in an event. **c** The IEA-loss imposes a pair-wise fine-grained class-to-class imitation force for the generator. Sg indicates that gradients are stopped for the discriminator, and only the generator's gradients will be updated.

improve the ability of the generator to generate more fine-grained samples in the simulation process by being aware of the variability of conditions at each event.

## IEA-GAN evaluation

Our study showcases the performance of IEA-GAN in generating ultra-high-granularity detector responses, demonstrated through its successful application to the ultra-high dimensional data of the Pixel Vertex Detector (PXD) at Belle II, consisting of over 7.5 million pixel channels. Furthermore, our findings reveal that the Fréchet inception distance (FID)[47] and Kernel Inception Distance (KID)[48] metrics for detector simulation are a very versatile estimator in conjunction with the marginal distributions and are associated with the other image level metrics. We show that by using IEA-GAN, we are able to capture the underlying distributions so that we can generate and amplify detector response information with a very good agreement with the Geant4 distributions. We also find out that the SOTA models in high-resolution image generation, even with an in-depth hyperparameter tuning analysis, do not perform well in comparison.

For evaluation, we have two categories of metrics: image level and physics level. As we are interested in having the best pixel-level properties, diversity, and correlation of the generated images

simultaneously while adhering to minimal generator complexity due to computational limitations, choosing the best iteration to compare results is challenging. Hence, we choose models' weights with the best FID for all comparisons. To compute the FID and KID scores, based on the recent Clean-FID project[49], we entirely fine-tuned the Inception-V3[50] model on the PXD images, as the PXD images are very different from the natural images used in their initial training. The downstream task for the fine-tuning was multi-class classification, involving 40 different sensors with which it acquired the ability to discriminate sensors. In other words, the classification task of the Inception-V3 involved identifying the specific sensor ID from a range of 1 to 40. This task required the model to identify the sensor to which each data sample belonged by discerning the data characteristics inherent to each sensor. This process can be done for any other detector dataset. FID measures the similarity of the generated samples' representations to those of samples from the real distribution. Given large sampling statistics, for each hidden activation of the Inception model, the FID evaluates the Fréchet distance, also known as Wasserstein-2 distance, between the first two moments of the activation distributions. As demonstrated to be useful and practical in the natural image analysis domain, FID performs[51] well in terms of discriminability, diversity, and robustness despite only modeling the first two moments of the

**Table 1 | FID and KID comparison between models (all models in the benchmark are highly tuned to the current problem and dataset), averaged across six random seeds (retrained and averaged across six models trained with different random seeds)**

|  | WGAN-gp | BigGAN-deep | ContraGAN | PE-GAN | IEA-GAN | Test Data |
|---|---|---|---|---|---|---|
| **FID** | 12.09 | 4.40 ± 0.88 | 3.14 ± 0.74 | 2.61 ± 0.91 | **1.50 ± 0.16** | $2.4 \times 10^{-5}$ |
| **KID**$^{(\times 10^{-3})}$ | 9.6 | 3.1 ± 0.1 | 1.5 ± 0.2 | 2.1 ± 0.4 | **1.0 ± 0.2** | $7.6 \times 10^{-1}$ |

The lower the FID and KID, the better the image quality and diversity. The bold values highlights the state of the art results by IEA-GAN.

distributions in the feature space. The lower the FID score, the more similar the distributions of the real and generated samples are. Kernel Inception Distance (KID) is another metric similar to FID, used for evaluating the quality of generative models. Unlike FID, KID uses a kernel two-sample test, which provides an unbiased estimate of the distance between distributions and is more robust to small sample sizes.

We compare IEA-GAN with three other models (only for image-level metrics) and the reference, which is the Geant4-simulated[52] dataset. The baselines are the SOTA in conditional image generation: BigGAN-deep[53] and ContraGAN[31]. We also compare IEA-GAN with the previous works on the PXD image generation task: PE-GAN[7] and WGAN-gp[6] (only for FID).

Table 1 demonstrates that generated images by IEA-GAN have the lowest FID and KID score compared to the other models and outperform them by 42%. This indicates that our model is able to generate synthetic samples that are much closer to the target data than the samples generated by the other models. The low FID and KID values for the Test Data indicate that the model has achieved a full understanding of the full data. In the Supplementary Table 3 in supplementary note, we demonstrate the sensitivity and possible interoperability of FID to various types of image distortions directly linked to the underlying physics recorded by the corresponding sensor. We achieved this by introducing controlled changes or jitters to the images and tracking their impact on the FID score.

At the pixel level, there are the pixel intensity distribution, occupancy distribution, and mean occupancy. The pixel intensity distribution defines the distribution of the energy of the background hits. The occupancy distribution and the pixel intensity distribution are evaluated over all sensors of a given number of events, while the mean occupancy corresponds to the mean value of sparsity across events for each sensor. This pixel-level information is essential since upon physics analysis via the basf2 software[54], when one wants to use the images and overlay the extracted information on the signal hits, the sparsity of the image defines the volume of the background hits on each sensor. The pixel intensity distribution, the occupancy distribution, as well as the mean occupancy per sensor are shown in Fig. 3. The distributions for the IEA-GAN model show the closest agreement with the reference.

The bimodal distribution of the occupancy comes from the geometry of the detector, as the sensors are not in a cylindrical shape like a calorimeter but in an annulus shape. This indicates how challenging generating this detector signature is concerning both its geometry and resolution. In order to capture the correct bi-modality of the occupancy distribution, the RRM and the Uniformity loss play an important role. By using the Uniformity loss in the discriminator, the generator is incentivized to produce samples that are not biased towards a particular mode or class, leading to a wider bimodal distribution of generated samples.

Moreover, by utilizing the RRM module that considers the interdependencies and correlations among the samples within an event, the IEA-GAN exhibits a superior consistency with high-energy hits, which enhances the diversity of generated samples in regions with lower occurrence rates.

Along with all these image-level metrics, we also need an intra-event sensitive metric. All the above metrics are equivariant under permutation between the samples among events. In other words, if we randomly shuffle the samples between events while fixing the sensor

number, all the discussed metrics are unchanged. Hence, we need a metric that looks at the context of each event individually in its event space and goes beyond the sample space. Ergo, we compute the Spearman's correlation between the occupancy of the sensors along the population of generated events,

$$\mathbf{r}_s = \mathrm{Corr}_p \left( R \left( \biguplus_{i=1}^{M=40} (\|\mathbf{x}_i\|_0) \right), R \left( \biguplus_{i=1}^{M=40} (\|\mathbf{x}_i\|_0) \right) \right), \quad (1)$$

where R(.) is the rank operator, a function that assigns a rank to each number in a list as in the definition of Spearman's correlation, and $\mathrm{Corr}_p$(.) is the Pearson Correlation function. $\biguplus$ is the disjoint union operator that symbolizes the concatenation operation. The norm with subscript 0, denoted by $\|.\|_0$, is the L0 measure. It is a function that counts the number of non-zero elements in a vector. In this work, it is used to calculate the occupancy of the sensors, i.e., the number of non-zero elements in the sensor image $x_i$. The coefficients by PE-GAN are random values in the range [−0.2, 0.2], whereas IEA-GAN images show a meaningful correlation among their generated images. Even though the desired correlation is different from the reference, as shown in Fig. 4, IEA-GAN understands a monotonically positive correlation for intra-layer sensors and a primarily negative correlation for inter-layer sensors.

In order to demonstrate that the learned correlation is actually meaningful, we incorporate the Mantel test[55,56], which is a significance test of the correlation between two distance/correlation matrices excluding the diagonal part. The Mantel test works by comparing each pair of corresponding elements in the two matrices. The null hypothesis is that there's no relationship between the two sets of correlations, and the test statistic is a correlation coefficient. The significance of the observed correlation is evaluated using permutation testing. This involves randomly rearranging the elements of one matrix many times, recalculating the test statistic each time, and then seeing how extreme the observed test statistic is relative to this null distribution of test statistics. If the observed test statistic is very extreme, then the *p*-value is <0.05, and the null hypothesis is rejected. For IEA-GAN, the Mantel test results show a veridical correlation of 18 ± 2% with empirical *p*-value of 0.0013. As the *p*-value is <0.05, we can reject the null hypothesis and observe that there is significant evidence for a correlation between the two sets of matrices. This suggests that the sensor classes that are more correlated in the Geant4 samples tend to also be correlated in the generated ones by IEA-GAN. Whereas for PE-GAN, the Mantel test results show a veridical correlation of 0.2% with empirical *p*-value of 0.96 in support of the null hypothesis.

While image level metrics indicate the low-level quality of simulations, we must also confirm that the resulting simulations are reasonable physics-wise when the entire detector is considered as a whole. For this, we do the tracking analysis to examine the Helix Parameter Resolutions (HPR). The quality of the tracking and HPRs directly impacts the precision and accuracy of the measurements.

At the Belle II experiment, after each collision event, tracks propagating in vacuum in a uniform magnetic field move roughly along a helix path described by the five helix parameters $\{d_0, z_0, \phi_0, \omega, \tan\lambda\}$ with respect to a pivot point[57]. The difference between the true and reconstructed helix parameters defines the resolution for the corresponding helix parameter. The track parameter resolution is affected by the number of hits, the hit intensity, and the underlying intra-event

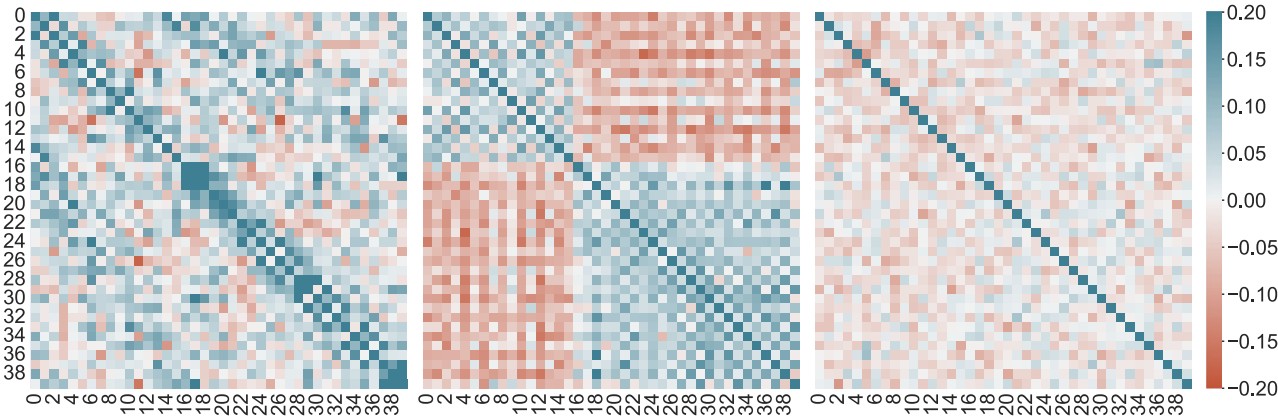

**Fig. 3 | Image-Level Histograms.** Pixel intensity distribution in linear (top left) and logarithmic scale (top right), the distribution of the occupancy (bottom left) and the mean occupancy per sensor (bottom right) for 10,000 events.

**Fig. 4 | Intra-Event Correlation.** Spearman's correlation between the occupancy of Geant4 sensor images (left), and sensor images from IEA-GAN (center), sensor images from PE-GAN (right).

correlation. Understanding how the background effects impact the HPR can give us crucial insights into the overall performance of the detector and the quality of the data it produces.

In this study, we utilize the same event generation and track reconstruction (employing the same set of signal events across all simulations to factor out signal fluctuations) for all simulations, ensuring that the signal hits across simulations are essentially identical. Consequently, the true track information remains consistent. Then, the primary point of difference lies in the origin of the background.

This distinct differentiation allows any disparities identified in the tracking parameter resolutions to be attributed largely to the different backgrounds and their generation origin, enabling a direct evaluation of the quality and performance of the IEA-GAN model in comparison to Geant4. It is essential to note that the core objective of this study is not to isolate or mitigate the effects of background noise but to simulate and measure its impact on track reconstruction efficiency accurately. The background affects the tracking to make it assign wrong hits. The background processes can create additional hits in the detector

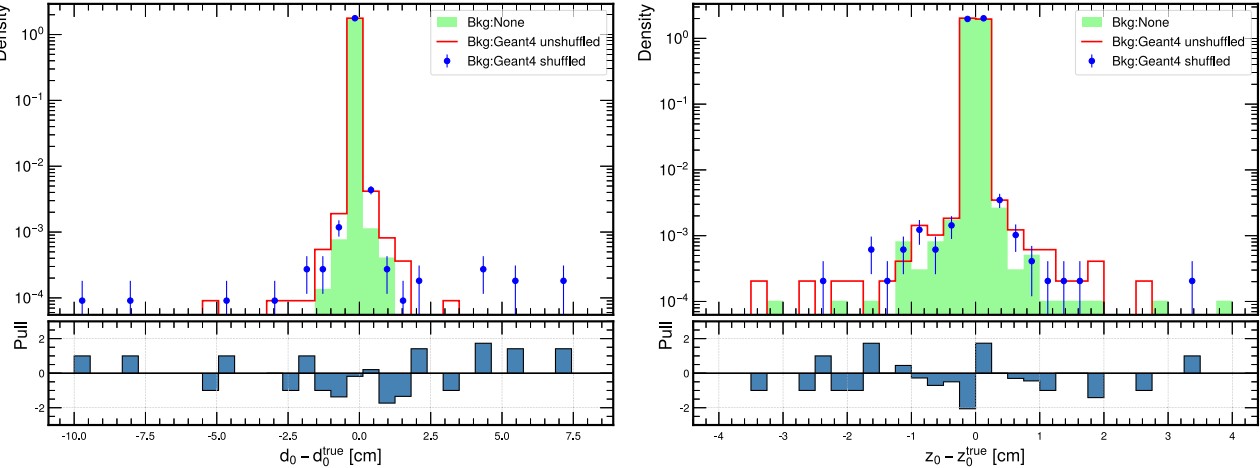

**Fig. 5 | The pull plot for the resolution of $d_0$ (left) and $z_0$ (right) in the presence of correlated (unshuffled) background and uncorrelated (shuffled) background at high momentum regime.** The pull, $\pi(\Delta(p))$ where p is a Helix parameter, is computed as $\pi(\Delta(p)) = \frac{\Delta p^{Shuff.} - \Delta p^{Unshuff.}}{\sqrt{\sigma^2_{\Delta p^{Shuff.}} + \sigma^2_{\Delta p^{Unshuff.}}}}$.

simulation that are not part of the actual particle trajectory. When these spurious hits are incorporated into the track reconstruction process, it leads to imperfect track parameter (Helix parameters) resolution. Thus, the effectiveness of the surrogate model is evaluated based on how well it replicates the Geant4 simulated background effects. As a result, we aim to ensure that IEA-GAN can generate the PXD background that impacts the track reconstruction process like Geant4-simulated background processes would.

First, as a physics motivation, we highlight the impact of the intra-event correlation by shuffling Geant4 samples. In other words, we show that in a physics analysis, the intra-event sensor-by-sensor correlation influences the performance of the tracking parameters. We examine the results by comparing the standard deviation of the Helix parameter resolutions and the 2-sample Kolmogorov-Smirnov test (KS test)[58] between the shuffled and unshuffled Geant4 PXD background. The standard deviation for each resolution, Δp, where p is a Helix parameter, is computed as follows:

$$\sigma_p = \sqrt{\frac{\sum_{i=1}^{n}(\Delta p^i - \overline{\Delta p})^2}{n}}, \qquad (2)$$

where $\sigma_p$ is the standard deviation for the Helix parameter p, $n$ is the number of the reconstructed tracks, and $\overline{\Delta p}$ is the resolution mean. The Error (margin of error) for the standard deviation (using the confidence interval method)[59,60] is

$$Error = \frac{\sqrt{\frac{n\sigma_p^2}{\chi^2_{1-\alpha/2,n}}} - \sqrt{\frac{n\sigma_p^2}{\chi^2_{\alpha/2,n}}}}{2}. \qquad (3)$$

**Table 2 | Standard deviation (using eq. (2) and eq. (3)) and KS test results for the shuffled and unshuffled Geant4 data across the 5 Helix parameters**

| Parameter | Standard deviation ± error | | KS statistic | *p*-value |
|---|---|---|---|---|
| | **Shuffled Geant4** | **Unshuffled Geant4** | | |
| $\Delta d_0$ [cm] | 0.1343 ± 0.0007 | 0.0732 ± 0.0004 | 0.0067 | 0.7655 |
| $\Delta \phi_0$ [rad] | 0.2158 ± 0.0011 | 0.1859 ± 0.0009 | 0.0066 | 0.7899 |
| $\Delta z_0$ [cm] | 5.0076 ± 0.0253 | 4.9341 ± 0.0249 | 0.0152 | 0.0211 |
| $\Delta \omega$ [cm⁻¹] | 0.0010 ± 0.0001 | 0.0008 ± 0.0001 | 0.0138 | 0.0485 |
| $\Delta \tan \lambda$ | 0.0388 ± 0.0002 | 0.0382 ± 0.0002 | 0.0167 | 0.0086 |

Where $\sigma_p$ is the standard deviation computed using eq. (2), $n$ is the number of the reconstructed tracks, $\chi^2_{\alpha/2,n}$ and $\chi^2_{1-\alpha/2,n}$ are the critical values from the chi-square distribution for $n$ degrees of freedom, and $\alpha$ is the significance level for a 95% confidence interval. For 5000 events, the results for the high momentum tracks, with more than 0.4 GeV show that there is strong evidence that losing the intra-event sensor-by-sensor correlation would impact the resolution and thus the precision of the $\mathbf{d}_0$, $\phi_0$ and $\omega$ Helix parameters. For the $\mathbf{z}_0$ and $\tan \lambda$ parameter, there is no significant difference in the standard deviation of the resolutions. However, the KS test for these parameters yields low *p*-values, indicating a high discrepancy between the shape of the two distributions.

In the context of each Helix parameter, for $\mathbf{d}_0$ impact parameter, the significant standard deviation in resolution shows that the loss of correlation directly impacts how well we can measure the particle's closest approach to the origin in the transverse plane. Losing sensor-by-sensor correlations that help to associate track hits correctly leads to a more spread out distribution of reconstructed values as shown in Fig. 5 and Table 2. This could affect subsequent analyses, such as identifying primary and secondary vertices, especially in scenarios where particles have negligible deflection (high momentum regime).

For the $\phi_0$ parameter, the insignificant resolution standard deviation difference and KS test result suggest that the lack of layer correlation doesn't significantly impact the distribution and precision of measurements of the azimuthal for high momentum tracks. The higher error in the standard deviation of $\Delta \mathbf{z}_0$ in the shuffled data suggests that the lack of correlation between detector layers introduces more uncertainty in determining the longitudinal interaction point. High momentum tracks are less likely to deviate significantly in the z-direction. Combined with the insignificant KS test result, this indicates a fundamental difference in how particle trajectories are reconstructed in the z-direction without layer correlation. $\omega$ is a measure of the curvature of the particle's track and is inversely proportional to the particle's momentum. For high-momentum particles, we would expect the curvature to be smaller since higher-momentum particles travel more linearly. The standard deviation for $\Delta \omega$ shows a slight discrepancy between the shuffled and unshuffled data, but the KS test doesn't show a significant difference. This indicates that while the overall distributions of the curvature resolution don't significantly differ, there's a minor difference in the precision with which the curvature is reconstructed, which could have implications for subsequent physics analyses that depend on accurate momentum information. Despite the insignificant resolution difference, the significant KS test

result for $\Delta \tan \lambda$ suggests differences in the inclination distributions of high momentum tracks between the shuffled and unshuffled data. This might indicate that the inclination of the track, which is also related to the momentum in the longitudinal direction, is affected by the loss of correlation between layers and sensors.

Now, we compare IEA-GAN with PE-GAN (the second-best performing model on the overall image level metrics) for the resolutions of all five helix parameters as shown in Fig. 6 and Table 3 for 5000 events for high momentum tracks ($P_T > 0.4$ GeV). In the low momentum region, the resolution performance of the models is on par. The tail behavior of $\Delta \phi$ comes from curling tracks where the direction of the tracks is swapped. Our meticulous comparison revealed that the standard deviation of these parameters, produced by the IEA-GAN model, approximates the Geant4 reference more closely, outperforming the PE-GAN model in each instance. Moreover, the Kolmogorov-Smirnov test results further consolidated our findings, showing higher $p$-values for the IEA-GAN model, thus adhering more accurately to the Geant4 reference. Another interesting observation is that, in comparison with the shuffled Geant4, IEA-GAN shows a more significant KS test $p$-value for $\mathbf{z}_0$, $\omega$, and $\tan \lambda$ resolutions and a more precise $\mathbf{d}_0$ reconstruction. Looking at the precision of IEA-GAN's $\mathbf{d}_0$ and $\mathbf{z}_0$ reconstruction, one can also deduce that despite only capturing a weak correlation, the downstream physics analysis, track reconstruction, benefits from even the weak correlation captured by IEA-GAN.

As a result of the analysis, we observe a good agreement between the IEA-GAN and Geant4, both in the tail segments (standard deviation) and precision of the resolutions where the most significant difference between Geant4 and no background is found. Hence, not only does IEA-GAN demonstrate a close image level agreement with Geant4, but it maintains a proper reconstructed physical behavior during track reconstruction as well.

## Discussion

In this work, we have proposed a series of robust methods for ultra-high-resolution, fine-grained, correlated detector response generation and conditional sampling tasks with our Intra-Event Aware GAN (IEA-GAN). IEA-GAN not only captures the dyadic class-to-class relations but also exhibits explainable intra-event correlations among the generated detector images while other models fail to capture any correlation. To achieve this, we present the Relational Reasoning Module (RRM) and the IEA-loss, with the Uniformity loss used in Deep Metric Learning. The RRM introduced a self-supervised relational contextual embedding for the samples in an event, which is compatible with GAN training policies, a task that is very challenging. It dynamically clusters the images in a collider event based on their inherent correlation, culminating in approximating a collision event. Our IEA-loss, a discriminator-supervised loss, helps the generator reach a consensus over the discriminator's dyadic relations between samples in each event. Finally, we have demonstrated that the Uniformity loss plays a crucial role for the discriminator in maximizing the homogeneity of the information entropy over the embedding space, thus helping the model overcome mode-collapse and capture a better bi-modality of generated occupancy.

As a result, an improvement to all metrics compared to the previous SOTA occurs, achieving an FID score of 1.50, an over 42% improvement, and a KID score of 0.0010, as presented in Table 1. Using IEA-GAN also comes with a storage release of more than 2 orders of magnitude. Furthermore, due to the dramatic CPU speed-up of ×147 as shown in Table 4, It is now possible to employ the IEA-GAN as an online surrogate model for the ultra high-granularity PXD background simulation on the fly, a task that was unattainable before for such a high-resolution detector simulation. Consequently, IEA-GAN, as a surrogate model that can generate more than 7.5M information channels, would be the applicable candidate that can handle the ultra-high granularity of the HL-LHC era. Moreover, we have shown that the application of the FID and KID metrics for the detector simulation provides a powerful tool for evaluating the performance of deep generative models in detector simulation. We have also illustrated the vital role that intra-event sensor-by-sensor correlation plays in downstream physics analysis. Consequently, we revealed that IEA-GAN, despite only capturing a weak correlation, surpasses PE-GAN and even outperforms the inter-event-shuffled (uncorrelated) Geant4 in certain metrics. We have also conducted an in-depth study into the optimal design and hyperparameters of the RRM, the IEA-loss, and the Uniformity loss. It is important to note that many existing models in calorimeter simulations do consider observables that depend on more than one layer simultaneously. These models typically focus on the simulation of particle showers from a single particle origin and a small region with the shower, which indeed capture aspects of inter-layer correlation within the scope of a localized area. However, our approach extends this concept by considering the entire event with multiple-particle origins that embrace the entire PXD detector as a whole, where correlations between different directions and depths (various angles and layers) become important within its readout window. Given the unique topology and geometry of PXD, this distinction is critical and allows for a more comprehensive full detector simulation, approximating the complex interplay within an event across the entire detector rather than just the localized particle shower. Furthermore, while previous studies may have implicitly accounted for layer-by-layer correlations within their framework, our study explicitly evaluates and compares these correlations and their influence on downstream physics analysis.

The ability to capture the underlying correlation structure of the data in particle physics experiments where the physical interpretation of the results heavily relies on it is very important. The true correlation between the occupancy of the sensors is determined by the underlying physical processes within the simulation. Although the actual correlation differs from the one captured by IEA-GAN, the model is learning patterns related to detector geometry, grounded in how these correlations manifest in the context of the PXD detector's structure. In particular, the model picked up a distinct positive layer-wise correlation, particularly between sensors 0–15 in the first layer and between sensors 16–39 in the second layer. This distinct pattern reflects a layer-wise understanding of the Toroidal geometry of PXD, although it differs partially from the actual correlations seen in the Geant4 data. This suggests that the difference in occupancy between inner and outer layers could be a major feature learned by the model, which may impede the learning of more subtle correlations. Therefore, while the IEA-GAN can provide valuable insights into the correlations and patterns present in the data, it is important to interpret its results in conjunction with the domain knowledge. To alleviate the discrepancy, we expect that incorporating perturbations directly into the discriminator's RRM module would improve its contextual understanding and, thus, the intra-event correlation. For example, using random masking[61] or inter-event permutation[62] over the samples and asking the RRM module to predict the representation of the perturbed sample will improve the robustness of the model.

This work significantly impacts high-granularity fast and efficient detector response and collider event simulations. Since they require fine-grained intra-event-correlated data generation, we believe that the Intra-Event Aware GAN (IEA-GAN) offers a robust controllable sampling for all particle physics experiments and simulations, such as detector simulation[11,63] and event generation[19,20,64,65] at both Belle II[37] and LHC[66]. In particular, the High-Luminosity Large Hadron Collider (HL-LHC)[25] is expected to surpass the LHC's design-integrated luminosity by increasing it by a factor of 10. For instance, the upcoming high-granularity Calorimeter (HGCAL) with roughly 6.5M channels, or the ITk 3D pixel detector at the HL-LHC[67] with around 1M information channels, will massively increase the geometry and precision

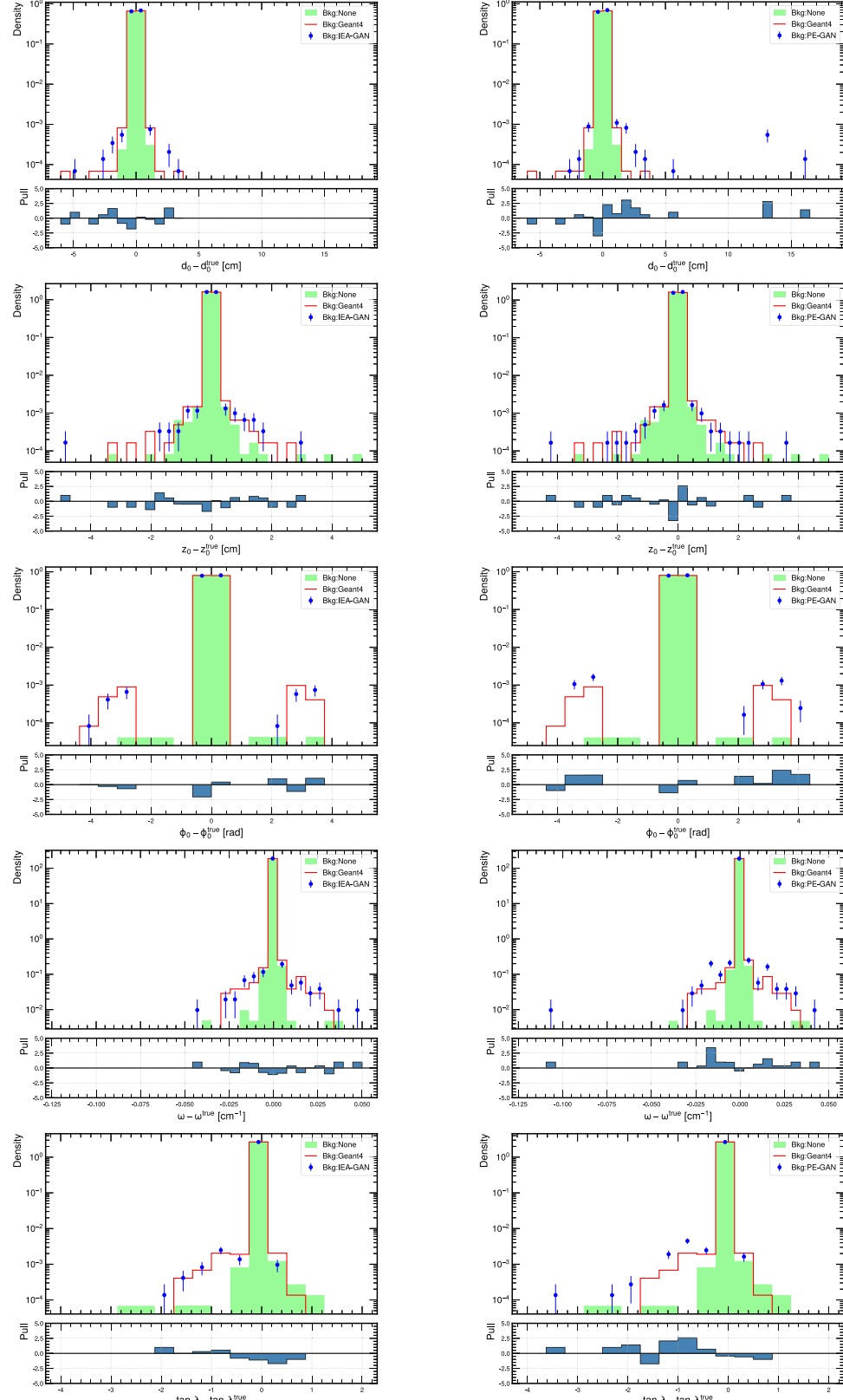

**Fig. 6 | The pull plots for comparing the Helix parameter resolutions for $d_0$, $z_0$, $\phi_0$, $\omega$, and $\tan\lambda$.** For each parameter, the left figure corresponds to the IEA-GAN simulated background and the right-side figure corresponds to the PE-GAN simulated background.

**Table 3 | Comparison of standard deviation and KS test results for the PE-GAN and IEA-GAN models with the Geant4 reference across 5 Helix parameters for high momentum tracks**

| Model | Parameter | Standard deviation ± error | | KS statistic | *p*-value |
|---|---|---|---|---|---|
| | | **Model** | **Geant4** | | |
| | $\Delta d_0$ [cm] | 0.1709 ± 0.0009 | 0.0732 ± 0.0004 | 0.0156 | 0.0164 |
| | $\Delta \phi_0$ [rad] | 0.2207 ± 0.0011 | 0.1859 ± 0.0009 | 0.0120 | 0.1193 |
| PEGAN | $\Delta z_0$ [cm] | 6.9073 ± 0.0349 | 4.9341 ± 0.0249 | 0.0183 | 0.0029 |
| | $\Delta \omega [\mathrm{cm}^{-1}]$ | 0.0014 ± 0.0001 | 0.0008 ± 0.0001 | 0.0116 | 0.1425 |
| | $\Delta \tan \lambda$ | 0.0579 ± 0.0003 | 0.0382 ± 0.0002 | 0.0179 | 0.0037 |
| | $\Delta d_0$ [cm] | 0.0762 ± 0.0004 | 0.0732 ± 0.0004 | 0.0104 | 0.2373 |
| | $\Delta \phi_0$ [rad] | 0.1905 ± 0.0010 | 0.1859 ± 0.0009 | 0.0109 | 0.1939 |
| IEA-GAN | $\Delta z_0$ [cm] | 5.1467 ± 0.0261 | 4.9341 ± 0.0249 | 0.0073 | 0.6814 |
| | $\Delta \omega [\mathrm{cm}^{-1}]$ | 0.0010 ± 0.0001 | 0.0008 ± 0.0001 | 0.0103 | 0.2537 |
| | $\Delta \tan \lambda$ | 0.0412 ± 0.0002 | 0.0382 ± 0.0002 | 0.0068 | 0.7538 |

**Table 4 | Computational performance of IEA-GAN and PE-GAN generators on a single core of an Intel Xeon Silver 4108 1.80GHz (CPU) and NVIDIA V100 with 32 GB of memory (GPU) compared to Geant4**

| Hardware | Simulator | Time/event [s] | Storage [Mb] | Speed-up |
|---|---|---|---|---|
| CPU | Geant4 | ≈1500 | ≈2000 | 1 |
| | PE-GAN | 11.781 ± 0.357 | ≈47 | ≈×127 |
| | IEA-GAN | 10.159 ± 0.208 | ≈47 | ≈×**147** |
| GPU | PE-GAN | 0.090 ± 0.010 | ≈47 | ≈×16,667 |
| | IEA-GAN | 0.070 ± 0.006 | ≈47 | ≈×21,429 |

For the generative models, the mean and standard deviation were obtained for sets of 10,000 events, meaning that the model generates these events one at a time, not in a batch of 10,000. The time for Geant4 refers to the theoretical time it would take to run the simulation of background processes on the fly, one event at a time. The storage consumption for Geant4 corresponds to storing 10,000 simulated events of 1 times the PXD background information, while for the surrogate models (i.e., IEA-GAN and PE-GAN), the term storage specifically refers to the models' weights. The bold values highlights the comutation gain of using IEA-GAN for surrogating the simulation.

complexity, leading to a dramatic increase in the time and storage to simulate the detector[68] As a result, much more effective and efficient high-resolution detector simulations are required. IEA-GAN is the potential candidate for simulating the corresponding full high-resolution and high-granular detector signatures with capability of generating more than 7.5M pixel channels. Nevertheless, while our approach offers a foundation for dealing with the complexities of any ultra-high granularity experiments, it is but a stepping stone. Our work aims to pave the way for such advancements, providing a preliminary framework capable of addressing the ultra-high granularity challenges.

Finally, IEA-GAN also has potential applications in protein design, which is a process that involves the generation of novel amino acid sequences to produce proteins with desired functions, such as enhanced stability and foldability, new binding specificity, or enzymatic activity[69]. Proteins can be grouped into different categories based on the arrangement and connectivity of their secondary structure features, such as alpha helices and beta sheets. Our developed intra-event aware methods, where an event represents the higher category of features, can also be applied to fine-grained density estimation[70] for generating new foldable protein[71–73] structures where category-level reasoning is of paramount importance.

## Methods
### Theory of generative adversarial networks
GANs are generative models that try to learn to generate the input distribution as faithfully as possible. For conditional GANs[74], the goal is to generate features given a label. Two player-based GAN models introduce a zero-sum game between two Synthetic Intelligent Agents, a generator network G, and a discriminator network D.

**Definition 4.1.** (Vanilla GAN) Given the generator G, a function $G : \mathbb{R}^d \to \mathbb{R}^n$, that maps a latent variable $\mathbf{z} \in \mathbb{R}^d$ sampled from a distribution to an n-dimensional observable, and the discriminator D, a functional $D : \mathbb{R}^n \to [0,1]$, that takes a generated image $\mathbf{x} \in \mathbb{R}^n$ and assigns a probability to it, they are the players of the following two-player minimax game with value function V(D, G)[9],

$$\min_G \max_D V(D,G) = \mathbb{E}_{\mathbf{x} \sim \mathbb{R}^n}[\log D(\mathbf{x})] + \mathbb{E}_{\mathbf{z} \sim \mathbb{R}^d}[\log(1 - D(G(\mathbf{z})))]. \quad (4)$$

After introducing the vanilla GAN, a vast amount of research has been undertaken to improve its convergence and stability, as, in general, training GANs is a highly brittle task. It requires a significant amount of hyperparameter tuning for domain-specific tasks. Many tricks, model add-ons, and structural changes have been introduced. A recent and comprehensive study prompted a very powerful SOTA model, BigGAN-deep[53], which incorporates the hinge-loss variation of the adversarial loss[75],

$$\mathcal{L}_D^{\mathrm{hinge}} = - \mathbb{E}_{\mathbf{x} \sim \mathbb{R}^n}[\min(0, -1 + D(\mathbf{x}))] - \mathbb{E}_{\mathbf{z} \sim \mathbb{R}^d}[\min(0, -1 - D(G(\mathbf{z})))], \quad (5)$$

$$\mathcal{L}_G^{\mathrm{hinge}} = - \mathbb{E}_{\mathbf{z} \sim \mathbb{R}^d}[D(G(\mathbf{z}))]. \quad (6)$$

Furthermore, many schemes for capturing the class conditions have been proposed since conditional GANs over image labels have been introduced[74]. The main idea is to minimize a specific metric between a class identification output of the discriminator and the actual labels after injecting an embedding of the conditional prior information into the generator. For example, ACGAN[76] tries to capture data-to-class relations by introducing an auxiliary classifier. The ProjGAN[30] also tries to capture these data-to-class relations by projecting the class embeddings onto the output of the discriminator via an inner product that contributes to the adversarial loss. The recent ContraGAN[31] incorporated concepts from metric learning or self-supervised learning (SSL) in order to seize data-to-data relations or intra-class relations by introducing the 2C loss, derived from NT-Xent loss[77], and proxy-based SSL,

$$\ell_{2C}(\mathbf{x}_i, \mathbf{y}_i) = - \log\left( \frac{\exp(S_c(\mathbf{h}(\mathbf{x}_i)^\top \mathbf{e}(\mathbf{y}_i))) + \sum_{k=1}^m \mathbf{1}_{k=i} \cdot \exp(S_c(\mathbf{h}(\mathbf{x}_i)^\top \mathbf{h}(\mathbf{x}_k)))}{\exp(S_c(\mathbf{h}(\mathbf{x}_i)^\top \mathbf{e}(\mathbf{y}_i))) + \sum_{k=1}^m \mathbf{1}_{k \neq i} \cdot \exp(S_c(\mathbf{h}(\mathbf{x}_i)^\top \mathbf{h}(\mathbf{x}_k)))} \right). \quad (7)$$

Here, $x_i \in \mathbf{x}$ are the images, $y_i \in \mathbf{y}$ are the corresponding labels, $S_c(.,.)$ is a similarity metric, $\mathbf{h}(.)$ is the output of the image embeddings, and $\mathbf{e}(.)$ is the output of the class embeddings. Although ContraGAN benefits from this loss by capturing the intra-class characteristics among the images that belong to the same class, it is prone to class-confusion[32,33] as different classes could also show similarity among themselves since their vector representation in the embedding space might not be orthogonal to each other, which is precisely what we are dealing with in a fine-grained dataset.

## Relational reasoning

Transformers[78] are widely used in different contexts. However, their application in Generative Adversarial Networks is either over the image manifold to learn long-range interactions between pixels[29,79] or via pure Vision-Transformer-based GANs[80] in which they utilize a fully Vision-Transformer[81] based generator and discriminator. Given the fact that training the Transformers is notoriously difficult[82] and task-agnostic when determining the best learning rate schedule, warm-up strategy, decay settings, and gradient clipping, fusing and adapting a Transformer encoder over a GAN learning regime is a highly non-trivial task. In this paper, we successfully merge a Transformer-based module adapted to the GAN training schemes for the discriminator's image and the generator's class modalities without any of the aforementioned problems.

**Definition 4.2.** (Attention) Transformers utilize a self-attention mechanism, the data of $(\mathbf{K}, \mathbf{Q}, \mathbf{V}, A)$. The vector spaces $\mathbf{K} \in \mathbb{R}^{N \times d_k}$, $\mathbf{Q} \in \mathbb{R}^{N \times d_k}$ and $\mathbf{V} \in \mathbb{R}^{N \times d_v}$ are the set of Keys, Queries, and Values. The bilinear map $a : \mathbf{K} \times \mathbf{Q} \to \mathbb{R}^{N \times N}$ is a similarity function between a key and a query. The attention, A, is defined as

$$A(\mathbf{K},\mathbf{Q},\mathbf{V}) := \text{Softmax}(a(\mathbf{K},\mathbf{Q}))\mathbf{V}, \tag{8}$$

where $d_k$ and $d_v$ are the dimensions of the corresponding vector spaces.

The attention mapping used in the vanilla Transformer[78] adopts the scaled dot-product as the bilinear map between keys and queries as

$$A(\mathbf{K},\mathbf{Q},\mathbf{V}) := \text{Softmax}\left(\frac{\mathbf{K}\mathbf{Q}^T}{\sqrt{d_k}}\right)\mathbf{V}. \tag{9}$$

The normalization factor $\frac{1}{\sqrt{d_k}}$ mitigates vanishing gradients for large inputs. Rather than simply computing the attention once, the multi-head mechanism runs through the scaled dot-product attention of linearly transformed versions of keys, queries, and values multiple times in parallel via learnable maps $\mathbf{W}_i^k$, $\mathbf{W}_i^q$ and $\mathbf{W}_i^v$. The independent attention outputs over $h$ number of heads are then aggregated and projected back into the desired number of dimensions via $\mathbf{W}^p$,

$$\text{MultiHead}(\mathbf{K},\mathbf{Q},\mathbf{V}) := \left[\biguplus_{i=1}^{h} \mathbf{H}_i\right]\mathbf{W}^p, \tag{10}$$

where $\mathbf{H}_i$ is given by $A(\mathbf{K}\mathbf{W}_i^k, \mathbf{Q}\mathbf{W}_i^q, \mathbf{V}\mathbf{W}_i^v)$. When used for processing sequences of tokens, the Self-Attention mechanism allows the transformer to figure out how important all other tokens in the sequence are, with respect to the target token, and then use these weights to build features of each token.

## Event Approximation.

An event, a single readout window after the collision of particles, consisted of 40 of images, each of which a sensor hitmap (image) of size $256 \times 768$. Thus, each event represents a round of detector signature collection. In order to approximate the concept of an event, at each iteration, IEA-GAN should take an event with 40 sensor images. Therefore, we are conditioning the model with the sensor type $[[1, 40]]$, which can be thought of as a mixture of angle and radius conditioning. These conditions have to enter the model as learnable *tokens* as they are not absolute and are context-based. It is impossible to pre-define meaningful sparse connections among the sample nodes in an event. For instance, the relation between images from different sensors can vary from event to event, albeit cumulatively, they follow a particular distribution. Ergo, the model has to learn any dynamical inherited conditions from the data in context (through the Relational Reasoning Module).

To model the context-based similarity between the different detector sensors in each event rather than their absolute properties, we have to use a permutation-equivariant[83,84] relational block that can encode pairwise correspondence among elements in the input set. For instance, Max-Pooling (e.g. DeepSets[85]) and Self-Attention[78] are the common permutation equivariant modules for set-based problems. Performing attention on all token pairs in a set to identify which pairs are the most interesting enables Transformers like Bert[45] to learn a context-specific syntax as the different heads in the multi-head attention might be looking at different syntactic properties[86,87].

Hence, we use a self-attention mechanism with weighted sum pooling as a form of information routing to process meaningful connections between elements in the input set and create an event graph. Each sample in an event is viewed as a node in a fully connected event graph, where the edges represent the learnable degree of similarity. Samples in each event go into message propagation steps of our Relational Reasoning Module (RRM), a GAN-compatible fully connected multi-head Graph Transformer Network[42–44].

**Relational reasoning module.** Specifically designed to be compatible with GAN training policies, the Relational Reasoning Module (RRM) can capture contextualized embeddings and cluster the image or class tokens in an event based on their inherent similarity.

Let $\mathbf{X} = \{\mathbf{x}_1, \ldots, \mathbf{x}_m\}$ be the set of the sampled images in each event, where $\mathbf{x}_i \in \mathbb{R}^d$, and $\mathbf{y} = \{\mathbf{y}_1, \ldots, \mathbf{y}_m\}$ be the set of labels, with $y_i \in [[1, 40]]$ for 40 detector (PXD) sensors. We also define two linear hypersphere projection diffeomorphisms, $\mathbf{h}_x : \mathbb{R}^k \to \mathbb{S}^n$ and $\mathbf{h}_y : \mathbb{Z} \to \mathbb{S}^n$, which map the image embedding manifold and the set of labels to a unit n-sphere, respectively. The unit n-sphere is the set of points, $\mathbb{S}^n = \{s \in \mathbb{R}^{n+1} | \, \| s \|_2 = 1\}$, that is always convex and connected. The Relational Reasoning Module benefits from a variant of the Pre-Norm Transformer[78] with a dot-product Multi-head Attention block such that,

$$\mathbf{p}_i'^{(l)} = \mathbf{p}_i^{(l)} + \sum_{k=1}^{h}\sum_{j=1}^{m} a_{ij}^{(l,k)} \mathbf{W}_{\text{SN}}^{(l)} \mathbf{LN}\left(\mathbf{p}_j^{(l)}\right), \tag{11}$$

$$\mathbf{p}_i^{(L)} = \mathbf{h}_x^{\text{LN}}\left(\mathbf{LN}\left(\bigcirc_{l=0}^{L}\left(\mathbf{p}_i'^{(l)} + \mathcal{F}_{\text{SN}}\left[\mathbf{LN}\left(\mathbf{p}_i'^{(l)}\right)\right]\right)\right)\right), \tag{12}$$

where $\mathbf{p}_i'^{(l)} \in \mathbb{R}^k$ is the embedding of each image via the discriminator for layer $l$ of the RRM. LN is the Layer Norm function[88] and $h$ is the number of heads defined in Eq. (10). $\mathcal{F}[\,.\,]$ is a two layer MLP functional defined as $\mathcal{F}_{\text{SN}}[\mathbf{p}_i^{(l)}] = \text{ReLU}(\mathbf{p}_i^{(l)}\mathbf{W}_{\text{SN}}^{(l,1)})\mathbf{W}_{\text{SN}}^{(l,2)}$ with Spectral Normalization[89]. The logits $a_{ij}^{(l,k)}$ are the normalized Attention weights of the bilinear function that monitor the dyadic interaction between image embeddings in layer $l$ and head $k$ defined in Eq. (9). $\mathbf{W}_{\text{SN}}^{(l)}$ in Eq. (11) is the learnable multi-head projector at layer $l$ defined in Eq. (10) with Spectral Normalization. The output of the composition of all layers via the composition of $L$ functionals, $\bigcirc_{l=0}^{L}\Phi^l := \phi_{w_L} \circ \ldots \phi_{w_0}[\mathbf{p}_i^{(l=0)}] \in \mathbb{R}^{m \times k}$, goes into a Layer Normalization layer where $\Phi^l = \mathbf{p}_i'^{(l)} + \mathcal{F}[\mathbf{LN}(\mathbf{p}_i'^{(l)})]$. $\mathbf{h}_x^{\text{LN}}(.)$ in Eq. (12) is the hypersphere compactification while the vectors are being standardized over the unit n-sphere $\mathbb{S}^n$ by a Layer Normalization.

For the discriminator, this module takes the set of image embeddings as input nodes within a fully connected event graph, applies a dot-product self-attention over them, and then updates each

sample or node's embedding via the attentive message passing, as shown on the right of Fig. 2. In the end, it compactifies the information by projecting the normalized graph onto a hypersphere via an L2 normalization[90]. Embedding the samples in an event on the unit hypersphere provides several benefits. In modern machine learning tasks such as face verification and face recognition[90], when dot products are omnipresent, fixed-norm vectors are known to increase training stability. In our case, this avoids gradient explosion in the discriminator. Furthermore, as $S^n$ is homeomorphic to the 1-point compactification of $\mathbb{R}^n$ when classes are densely grouped on the n-sphere as a compact convex manifold, they are linearly separable, which is not the case for the Euclidean space[91].

For the generator's RRM, we use a simpler version of the above dot-product Multi-head Attention block without the last hypersphere compactification due to the stability issues, as shown on the left of Fig. 2. It finds a learnable contextual embedding for each event that will be fused to each class token via the feature mixing layer, which is a matrix factorization linear layer $\mathbf{W}_{SN}(.)$. Formally, we have,

$$\mathbf{q}_i^{(0)} = \mathbf{W}_{SN}\left(\mathbf{r}_i \uplus \mathbf{e}_i\right), \tag{13}$$

$$\mathbf{q}_i'^{(l)} = \mathbf{q}_i^{(l)} + \sum_{k=1}^{M}\sum_{j=1}^{m} a_{ij}^{(l,k)} \mathbf{W}^{(l)} \mathbf{LN}\left(\mathbf{q}_j^{(l)}\right), \tag{14}$$

$$\mathbf{q}_i^L = \mathbf{LN}\left(\bigcirc_{l=0}^{L}\left(\mathbf{q}_i'^{(l)} + \mathcal{F}\left[\mathbf{LN}\left(\mathbf{q}_i'^{(l)}\right)\right]\right)\right), \tag{15}$$

where $\mathbf{e}_i : \mathbb{Z} \to \mathbb{R}^t$ is the embedding of each class token via the embedding layer of the generator. The logits $a_{ij}^{(l,k)}$ are the normalized Attention weights of the bilinear function that monitor the dyadic interaction between classes in the event embeddings in layer $l$ and head $k$ defined in Eq. (9). $\mathbf{W}^{(l)}$ in Eq. (14) is the learnable multi-head projector at layer $l$ defined in Eq. (10). The output of the composition of all layers via the composition of $L$ functionals, $\bigcirc_{l=0}^{L}\Phi^l := \phi_{w_L} \circ ... \phi_{w_0}[\mathbf{q}_i^{(l=0)}] \in \mathbb{R}^{m \times t}$, goes into a Layer Normalization layer where $\Phi^l = \mathbf{q}_i'^{(l)} + \mathcal{F}[\mathbf{LN}(\mathbf{q}_i'^{(l)})]$ as shown in Eq. (15).

One input to the generator is the embedded labels, which can be considered rigid token embeddings that will be learned as a global representation bias of each sensor. As sensor conditions change for each event as a set, having merely class embeddings, as used in conditional GANs[74], is insufficient because the context-based information will not be learned. Thus, the generator samples from a per-event shared distribution at each event as random degrees of freedom (Rdof). Rdofs are random samples from a shared Normal distribution for each class, $\mathbf{r}_i \sim \mathcal{N}(0,1)$, that introduces four-dimensional learnable degrees of freedom for the generator, see Eq. (13) This way, we ensure that the generator is aware of intra-event local changes, culminating in having an intra-event correlation among the generated images. Rdof can be interpreted as both perturbation[46] to the token embeddings and an event-level segment embedding[45], which can enhance the diversity of the generated images.

## Intra-event aware loss

Motivated by Self-Supervised Learning[92], to transfer the intra-event contextualized knowledge of the discriminator to the generator in an explicit way, we introduce an Intra-Event Aware (IEA) loss for the generator that captures class-to-class relations,

$$\ell_{IEA}(\mathbf{x}_r, \mathbf{x}_f) = \sum_{i,j} D_{KL}\left(\sigma\left(\mathbf{h}\left(\mathbf{x}_i^{(r)}\right)^\top \mathbf{h}\left(\mathbf{x}_j^{(r)}\right)\right) \middle\| \sigma\left(\mathbf{h}\left(\mathbf{x}_i^{(f)}\right)^\top \mathbf{h}\left(\mathbf{x}_j^{(f)}\right)\right)\right), \tag{16}$$

where $\mathbf{x}_r = \{\mathbf{x}_i^{(r)}\}_{i=1}^m$ is the set of real images, and $\mathbf{x}_f = \{G(\mathbf{z}^i, \mathbf{y}^i, \mathbf{r}^i) = \mathbf{x}_i^{(f)}\}_{i=1}^m$ the set of generated images. The softmax function, $\sigma : \mathbb{R}^m \to [0,1]^m$, normalizes the dot-product self-attention between the image embeddings. The map $\mathbf{h} : \mathbb{R}^k \to \mathbb{S}^n$ is the unit hypersphere projection of the discriminator. Therefore, the dot product is equivalent to the cosine distance. $D_{KL}(. \| .)$ is the Kullback-Leibler (KL) divergence[93] which takes two $m \times m$ matrices that have values in the closed unit interval (due to the softmax function). Hence, having a KL divergence is natural here as we want to compare one probability density with another in an event. We also tested other distance functions reported in the supplementary note. By considering the linear interaction[34] between every sample in an event and assigning a weight to their similarity, the generator mimics the fine-grained class-to-class relations within each event and incorporates this information in its RRM module as shown in Fig. 2.

Upon minimizing it for the generator (having the stop-gradient for the discriminator), we are putting a discriminator-supervised penalizing system over the intra-event awareness of the generator by encouraging it to look for more detailed dyadic connections among the images and be sensitive to even slight differences. Ultimately, we want to maximize the consensus of data points on two unit hyperspheres of real images and generated image embeddings.

## Uniformity loss

The other crucial loss function comes from contrastive representation learning. With the task of learning fine-grained class-to-class relations among the images, we also want to ensure the feature vectors have as much hyperspherical diversity as possible. Thus, by imposing a uniformity condition over the feature vectors on the unit hypersphere, they preserve as much information as possible since the uniform distribution carries a high entropy. This idea stems from the Thomson problem[94], where a static equilibrium with minimal potential energy is sought to distribute N electrons on the unit sphere in the evenest manner. To do that, we incorporate the uniformity metric[35], which is based on a Gaussian potential kernel,

$$\mathcal{L}_{uniform}(\mathbf{x}; s) = \log \mathbb{E}_{\mathbf{x}_i, \mathbf{x}_j \sim p_{event}}\left[\exp\left(s \| \mathbf{h}(\mathbf{x}_i) - \mathbf{h}(\mathbf{x}_j)\|_2^2\right)\right]. \tag{17}$$

Upon minimizing this loss for the discriminator, it tries to maintain a uniform distance among the samples that are not well-clustered and thus not similar. In other words, eventually, we want to reach a maximum geodesic separation incorporating the Riesz s-kernel[95] with $s = -2$ as a measure of geodesic similarity, to preserve maximal information over the Hypersphere. Therefore, asymptotically it corresponds to the uniform distribution on the hypersphere[96]. This loss is beneficial for capturing the exact distribution of the mean occupancy distribution and balancing the inter-class pulling force of the Relational Reasoning module. As a result, not only does it help generate more diverse and varied outputs, but it also can prevent issues such as mode collapse or overfitting.

## Model details and hyperparameters

In this study, we utilized a dataset of 40,000 Monte Carlo simulated events[40], of which 35000 were allocated for training, 5000 for model selection (validation), and an independent set of 10000 events served as the test set for assessing the final model performance. It is noteworthy to acknowledge that this is a rather small dataset to train a deep generative model for $\mathcal{O}(10^7)$ data channels. The data in each event consists of 40 grey-scale $256 \times 768$ zero-padded images. They are zero-padded on both sides from their original size of $250 \times 768$ to be divisible by 16 for training purposes.

To capture the intra-event mutual information among the images using the RRM and approximate the concept of an event, the model

samples (and generates) an entire event at each iteration. This approach ensures that each event in our analysis comprises a correlated set of 40 unique images. All hyperparameters are chosen based on the model's stability and performance upon the validation set. The learning rates for the Generator and Discriminator are $5 \times 10^{-5}$ with one sample per class sampler. The Relational Reasoning Module of the Generator has two heads and one layer of non-spectrally normalized message propagation with an embedding dimension of 128 and ReLU non-linearity. The input to the generator's RRM is embedded class tokens mixed with 4 random degrees of freedom by a spectrally normalized linear layer.

For the Discriminator, the RRM has four heads with one layer of spectrally normalized message propagation with the embedding dimension 1024 as the hypersphere dimension and ReLU non-linearity. All Generator and Discriminator modules use Orthogonal initialization[97]. For the IEA-loss in Eq. (16), the coefficient $\lambda_{IEA} = 1.0$ (highlighted in Supplementary Algorithm 1 in supplementary note) gives the best result. The most stable contribution of the Uniformity loss, defined in Eq. (17), is with $\lambda_{uniform} = 0.1$. For the backbone of both the discriminator and the generator, we use BigGAN-deep[53] with a non-local block at channel 32 for the discriminator only. Since there is no meaningful way to define a minimal loss in GAN training, our stopping point is the divergence of the FID.

## Data availability
The data used in this study is openly available at https://zenodo.org/record/8331919[98].

## Code availability
The code for this study is available at https://github.com/Hosein47/IEA-GAN[99].

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

## Acknowledgements

This work was part of B.H's PhD thesis, done while at Ludwig Maximilians University in Munich. This research was supported by the collaborative project IDT-UM (Innovative Digitale Technologien zur Erforschung von Universum und Materie) and KISS consortium funded by the German Federal Ministry of Education and Research (BMBF) and the Deutsche Forschungsgemeinschaft under Germany's Excellence Strategy - EXC 2094 ORIGINS - 390783311. JK's work is supported by the Helmholtz Association Initiative and Networking Fund under the Helmholtz AI platform grant. B.H. and S.S. wish to express their gratitude to Volker Tresp and Thomas Lueck for their valuable discussions that enriched this work. We also thank David Katheder for his assistance with preparing the GitHub repository. We thank our colleagues from the Ludwig Maximilian University in Munich and the Computational Center for Particle and Astrophysics (C2PAP), who provided expertize and computation power that greatly assisted the research.

## Author contributions

B.H. designed and implemented the main idea of the research, conducted all the experimental runs, performed the downstream physics analysis, and wrote the manuscript. N.H., S.S., and T.K. contributed to maturing the research idea at various stages. N.H. contributed to several parts of the code, plots, data analysis pipeline, and validation of the

results. T.K. guided the physics analysis and validation of the results. S.S. steered the idea Graph Transformer and relational reasoning. J.K. guided and conducted the development of the manuscript and plots at all stages. All authors have reviewed and commented on the manuscript.

## Funding

## Competing interests
The authors declare no competing interests.
