## [Peer Review File · Nature Communications]

Ultra-High-Granularity Detector Simulation with Intra-Event Aware Generative Adversarial Network and Self-Supervised Relational ReasoningREVIEWER COMMENTS

Reviewer #1 (Remarks to the Author):

The author introduced novel machine learning techniques to generate “contextualized high-resolution detector responses.” that capture the intra-event correlations. Those techniques include the Relational Reasoning Module, intra-event aware loss, and Uniformity loss. The proposed model, namely IEA-GAN, is compared with the PE-GAN.

One major issue compels me not to recommend the paper for publication in this journal.

There is no physics motivation for modeling the intra-event level correlations. On the contrary, the author demonstrated that a model that cannot describe the intra-event level correlations could result in a good agreement in tracking parameter resolutions. Such models are IEA-GAN and PE-GAN. Even if the intra-event correlation was important for some physics analyses, the IEA-GAN could model the correlation, as shown in Figure 3.

In addition, the author claims, “Furthermore, for the first time, our findings reveal that the FID [33] metric for detector simulation is a very versatile and accurate unbiased estimator in comparison to marginal distributions, and is correlated with both high and low-level metrics.” I failed to find any evidence in this paper to support that statement. On the contrary, a theoretical study in Ref[1] shows the FID is biased.

[1] <https://openreview.net/forum?id=r1IUOzWCW>

Reviewer #2 (Remarks to the Author):

In this manuscript, the authors present a Generative Adversarial Network (GAN)-based neural network they refer to as IEA-GAN which designed to generate contextualized simulated data applied to high-resolution detector responses. They include several novel elements in this generative model such as intra-event aware and uniformity loss elements and a relational reasoning module to approximate events at detectors, applied to the specific

case of the Belle-II experiment's high-granularity pixel vertex detector (PXD).

GANs are notoriously unweildy for this type of application, especially when employed to generate realistic data involving highly granular instrumentation designed to detect and measure particles at colliders or other other scientific experiments with high resolution. The authors optimizations and additions to the traditional GAN setup are well-motivated to designed address these challenges as described in detail in the manuscript with sufficient detail to understand and possibly reproduce the setup and results.

In regard to reproducibility of their results, the authors mention a forthcoming publically-available open dataset which this reviewer views to be the minimum required for publication. The authors are strongly encouraged to also ensure the data samples upon which is manuscript is based will adhere to the Findable, Accessible, Interoperable and Reusable (FAIR) principles, for which the authors might take as guidance the process detailed in <https://arxiv.org/abs/2108.02214>. Additionally for reproducibility, the authors are encouraged to make available example code for their model implementation on a public repository such as Github.

The work by the authors has the potential to be of great significance to the field of partical physics and related fields. The authors allude to this potential impact in the manuscript and this reviewer agrees with their assessment. For example, the experiments at CERN's Large Hadron Collider spend a large fraction of their computing resource on Geant4 simulation of collision events. If a robust solution to this challenge can be found and deployed, it will dramatically reduce the CPU required to do the LHC science by speeding up the event simulation by orders of magnitude, especially for interpreation of data during the HL-LHC era in which 10-20 times more collision data will be acquired as compared to previous LHC runs.

Regarding the reduction in resources via IEA-GAN approach to simulation, the authors emphasize the storage savings more so than the CPU savings. For example, early in the manuscript (on page 3), they specifically call out a factor of 2 reduction in storage requirements for pre-produced backgrounds for the PXD in Belle II simulated data but make

no mention of the CPU reductions there (they are shown later in Table 2). It is worth stressing that the potential benefit of ML-based fast simulation for the (HL-)LHC and other data-intensive science experiments is mostly in the CPU reduction, however the storage reduction associated with on-the-fly generation of backgrounds for overlay is an important benefit as well. It is suggested that that discussion on page 3 be adjusted to put the CPU and storage in benefits in proper context and at least equal footing.

The description in the introduction section is solely focused on GANs. The paper would be strengthened with a more general introduction to fast generative models for science, particularly including a mention of variational autoencoders (VAEs) which have proven to be at least as effective for fast event generation as GANs. For example, see references <https://arxiv.org/abs/2203.00520> and <https://arxiv.org/abs/1901.00875>.

The authors claim that the IEA-GAN study described in the manuscript with the Belle-II PXD is the "highest spatial resolution detector simulation ever analyzed with deep generative models". I could not find a specific published counter-example to this rather bold statement, however there have been GAN-based generative model studies with, for example, the geometry of the proposed CMS High Granularity Calorimeter (HGCal) detector upgrade for the HL-LHC. The authors are encouraged to do a thorough literature search to ensure this claim is correct.

Q: Does the work support the conclusions and claims, or is additional evidence needed?

There are several issues with the description, and possibly the methodology, used in paper in regard to the key results and their interpretation.

The primary weakness in the paper is a lack of evidentiary support for the two primary claims made in the paper (Section 2.2): (1) IEA-GAN "outperforms the other models considered by the authors by a wide margin" and (2) the authors' findings "for the first time, reveal that the FID metric for detector simulation is a very versatile and accurate unbiased estimator in comparison with marginal distributions, and is correlated with both high and low level metrics."

The authors use the FID metric to support their claim that IEA-GAN is superior to the other method considered but at the same time the authors claim their work supports FID as a metric of superiority. This is a circular argument.

Additionally, to use the FID metric results shown in Table 1 as a convincing argument in support of the conclusion that IEA-GAN is superior "by wide margin", there should be some validation of the FID metric itself presented in the manuscript. It is well known that the interpretation of ML model evaluation tools and metrics can depend on the specific application and not be fully general and domain agnostic. This is too important of an assumption to leave solely to a reference on the FID metric and therefore more details should be provided in the manuscript and/or additional studies to validate the FID metric as a means of assessing performance superiority of different generative models.

The studies with the FID metric are what I believe the authors refer to as "low-level" metrics.

For the "high-level physics" metric of comparing IEA-GAN tracking parameter resolutions against Geant, this is a necessary but insufficient study to claim to validate the conclusions of the FID metric results and general quality of the IEA-GAN generative model. Tracking resolutions, and therefore comparisons between two event sets, depend critically on the details of the hits that are used to reconstruct the tracks. E.g. How similar are the number of hits and hit positions between two simulated sets on average and track-by-track? More information such as these are required to make an apple-to-apple comparison of track parameter resolutions and support the authors interpretation of their results as a validation of the agreement between IEA-GAN and Geant simulations.

A few final comments on style of text in the manuscript. There are many places where capitalization is used on terms in a strange and inconsistent manner. For example, in the abstract it is written "Uniformity loss" but on page 7 it is written "uniformity loss". Also, there are occasions where symbols common in machine learning (ML) and data science (DS) communities but not as much in other scientific domains are used without definition. The manuscript would be more accessible to a broader audience if the authors were more

liberal with symbol definitions in their technical descriptions even if they are "obvious" within ML/DS communities.

Reviewer #3 (Remarks to the Author):

Dear Authors, dear Editor,

Report on the Manuscript "Ultra-High-Resolution Detector Simulation with Intra-Event Aware Generative Adversarial Network and Self-Supervised Relational Reasoning" by Hosein Hashemi, Nikolai Hartmann, Sahand Sharifzadeh, James Kahn and Thomas Kuhr for Nature Communications.

The manuscript describes a new algorithm to simulate the detector response of the Belle II Pixel Vertex Detector (PXD) using a deep generative model. In detail, the authors introduce the "Intra-Event Generative Adversarial Network" (IEA-GAN), which uses several innovative components to generate events in more than 7.5M channels. To my knowledge, it is the first algorithm to tackle dimensionalities beyond $1e6$, which is at least one (probably two) orders of magnitude above the biggest generative models for detector simulation in HEP.

The results are impressive, significantly better than other approaches, and definitively deserve to be published. However, I don't think the current version can be published in Nature Communications.

First, even though the presented algorithm shows a much better performance than any other considered deep generative algorithm, the problem of learning the underlying probability density as induced by GEANT4 is far from solved. Figure 2 still shows substantial differences between the generated distributions and the GEANT4 distribution, and it only shows a few different 1-dim projections of the 7.5M-dimensional space. I guess other observables also deviate, as well as the correlation between them. The fact that correlations are not learned properly is also shown in Figure 3 for a subset of observables.

This does not diminish the great achievement of managing to train a 7.5M-dimensional deep generative model (again, these results should be published), but I'm not sure it qualifies a

publication in the Nature portfolio, which, according to the homepage "should represent an advance in understanding likely to influence thinking in the field, [...]." In addition, "There should be a discernible reason why the work deserves the visibility of publication in a Nature Portfolio journal [...]" and I'm not sure this is the case here (but I'm happy to hear arguments for that).

Second, the way the manuscript is currently structured makes it nearly impossible to read and understand. I personally don't understand why it is journal policy to have the Methods section at the end and not use IMRaD (as most other journals publishing HEP research). This way, the reader has to flip back and forth through many pages in order to understand what is going on in the results section. Given that this can probably not be changed, there is still a lot that can be improved. For example, the description of the dataset and the general problem (2nd paragraph of section 4.1) could all go to the introduction. Reading that section, it is still unclear to me if the algorithm is supposed to be used for the simulation of the entire event, or only the background (cf Fig. 5d). Further, it is never explained what "background" refers to in that context (and what signal, if at all, was used in the dataset).

Third, there are many words used in the manuscript that (for me) mean the same things, but apparently are not the same things. For example why should there be more than one sample in one event? Isn't a sample from the generative model an event in the detector? Are "images" referring to events, too? Or only to single detector elements? Are these the same as samples? What are the "classes" that they authors refer to? Usual conditional deep generative models have different particle types, or incident angles, or incident energies as (class) condition, but none of these apply here. What are "tokens"?

Forth, there are many minor details that are either missing or not well explained. For example:

- Figure 1 (which is not the first figure being referred to) shows the architecture of the IEA-GAN. In many places, it is said that it shows how event graphs are built (in section 4.3.2) and how they are projected into a hypersphere. If at all, it shows that these steps are part of the algorithm, but not how these graphs look like, how they are constructed, or how such a spherical projection is achieved.

- Equations, like the one for r_s , are not sufficiently explained. What's a Rank variable? What is the norm with subscript 0? What does the union with + mean? What is e in I_{2C} ?
- Did the authors check that the Gaussian assumption to obtain the FID score can be made and that the distributions are actually Gaussian?
- Figure 3 left and center look very different. How do the authors conclude that the IEA-GAN captures the correlations?
- Are the 5 physical parameters of Figure 4 the only 5 relevant physics parameters that are extracted from the event? How do correlations between them look like? How do distributions of other quantities look like?
- The computational times in Table 2 show averages over 10000 generated events. Does that mean the GAN was run with a batchsize of 10000? Is GEANT4 (or the simulation chain as a whole) batched in Belle II? If not, only comparisons with batchsize 1 are meaningful.
- Was the ablation study of Appendix A performed in a grid, with all parameters being varied at the same time? Or only one parameter at a time?
- Appendix B, called "Extended Figures and Tables" only contains a single figure and no table and no explaining text. Is that intentional by the authors or was there anything cut in processing?

Response to Reviewers

Manuscript Title: Ultra-High-Resolution Detector Simulation with Intra-Event Aware Generative Adversarial Network and Self-Supervised Relational Reasoning

Introduction

We appreciate the thoughtful and inspirational comments of the reviewers on our manuscript, and we thank them for their time in providing this feedback. We have revised our manuscript with care to address these comments. Below, we respond to each comment in detail.

Response to Reviewer 1

Comment 1.1 There is no physics motivation for modeling the intra-event level correlations. On the contrary, the author demonstrated that a model that cannot describe the intra-event level correlations could result in a good agreement in tracking parameter resolutions. Such models are IEA-GAN and PE-GAN.

Response: We agree that a clear motivation was missing in the previous version of the paper. As a result of the reviewer’s instructive comment, we studied the Helix parameter resolutions for the simulated events in more details. We compared these results with a shuffled version of the events, where the data of each sensor was taken from a random event such that the only difference is the absence of correlation.

Our examination involved comparing the unbiased resolution and performing a 2-sample Kolmogorov–Smirnov test (KS test) on the Helix parameters between the shuffled and unshuffled Geant4 PXD background. The results, particularly for high momentum tracks, shown in Table 2 and Figure 5, provide evidence that the loss of intra-event sensor-by-sensor correlation can affect the resolution of the \mathbf{d}_0 , ϕ_0 and ω helix parameters. For the z_0 and $\tan \lambda$ parameters, while there was no significant difference in resolution, the KS test yielded low p-values, indicating a high discrepancy between the shapes of the two distributions. Thus, accordingly we updated the manuscript with further discussions in lines 360 – 405, under section 2.2.

Comment 1.2 On the contrary, the author demonstrated that a model that cannot describe the intra-event level correlations could result in a good agreement in tracking parameter resolutions. Such models are IEA-GAN and PE-GAN.

Response: Indeed, our initial assertion that PE-GAN, which does not capture intra-event level correlations at all, could still result in a good

agreement in tracking parameter resolutions, was not a complete statement and is partly true. Your feedback has encouraged us to delve deeper into this matter, prompting a more nuanced examination of the differences.

We conducted a detailed comparison of IEA-GAN and PE-GAN with Geant4 simulated data for the resolutions of all five helix parameters for high momentum tracks ($P_T > 0.4 \text{ GeV}$), as detailed in table 3 and depicted in figure 6 of the paper. We found that in the low momentum region, the resolution performance of the models are comparable. However, for high momentum tracks, our analysis revealed that the even the weakly captured intra-event correlation by IEA-GAN plays a role in tracking, which aligns with our earlier findings from the shuffling test analysis.

Upon this detailed analysis, we revise our earlier statement that PE-GAN is in good agreement in tracking parameter resolutions. This claim holds partially true only in the low momentum region. For high momentum tracks ($P_T > 0.4 \text{ GeV}$), our meticulous comparison revealed that the unbiased variance of the resolution of these parameters, produced by the IEA-GAN model, approximates more closely to the Geant4 reference, outperforming the PE-GAN model in each instance. The manuscript has been updated to reflect these findings, specifically in lines 405 – 420 and showed the results in table 3 and figure 6.

Comment 1.3 Even if the intra-event correlation was important for some physics analyses, the IEA-GAN could not model the correlation, as shown in Figure 3.

Response: We concur that the IEA-GAN model has not perfectly captured the correlation. However, we argue that the correlation it has learned, albeit weak, holds significant value. As the first work in fast detector simulation that considered such inter-sensor/layer correlation, we view this as a substantial advancement, despite the limitations indicated.

To establish the relevance of this learned correlation, we employed the Mantel test, a statistical method specifically designed to assess the association between two distance matrices while excluding the diagonal part. When we applied the Mantel test to IEA-GAN, the outcome demonstrated a veridical correlation of 0.18 ± 0.02 with empirical p-value 0.0013. Given that the p-value is below the typical threshold of 0.05, this permits us to reject the null hypothesis and to acknowledge a significant evidence for correlation between the two sets of matrices. This suggests that the sensor classes that are more correlated in the Geant4 samples tend to also be correlated in the generated ones by IEA-GAN. Whereas for PE-GAN the Mantel test results shows a veridical correlation of 0.002 with empirical p-value 0.96.

Moreover, we demonstrated that the reconstruction of the \mathbf{d}_0 impact parameter is dependent on the intra-event correlation. IEA-GAN, despite only capturing a weak correlation, showed a close agreement in track reconstruction with the correlated Geant4. This finding further

underscores the usefulness and benefits of the correlation captured by IEA-GAN.

In summary, although there is room for further optimization and enhancement of the IEA-GAN’s performance in modeling intra-event correlations, we believe this work represents a significant step towards this goal. We updated the manuscript with more details of the above analyses and methodology mainly in lines 318 – 337.

Comment 1.4 In addition, the author claims, Furthermore, for the first time, our findings reveal that the FID metric for detector simulation is a very versatile and accurate unbiased estimator in comparison to marginal distributions, and is correlated with both high and low-level metrics. I failed to find any evidence in this paper to support that statement. On the contrary, a theoretical study in [1] shows the FID is biased.

Response:We appreciate your insightful comment and agree with your observation. We apologize for our previous overstatement and would like to clarify our position. We now assert that the FID metric for detector simulation is a versatile estimator *in conjunction* with the marginal distributions. As demonstrated to be useful and practical in the natural image analysis domain, FID performs [3] well in terms of discriminability, diversity, and robustness, despite only modeling the first two moments of the distributions in the feature space. However, as you rightly pointed out, it is indeed a biased estimator. Thus, motivated by your suggestion, we have also incorporated the KID score [1] as an unbiased estimator. We now report both FID and KID scores in our comparisons, as shown in table 1, and have updated section 2.2 with the new information

Regarding the correlation of FID with image-level distributions, we conducted a sensitivity analysis of FID to various types of image distortions linked to the signatures recorded by the corresponding sensor. We introduced controlled changes or ‘jitters’ to the images and tracked their impact on the FID score, as presented in the table B3 of the revised manuscript. Therefore, we now suggest that FID and other metrics could be interrelated, and one can use FID as a data-centric metric for the initial evaluation of the model in the data domain of detector simulation. We updated the disclaimer in section 2.2 accordingly.

Response to Reviewer 2

Comment 2.1 In regard to reproducibility of their results, the authors mention a forthcoming publicly available open dataset which this reviewer views to be the minimum required for publication. The authors are strongly encouraged to also ensure the data samples upon which is manuscript is based will adhere to the Findable, Accessible, Interoperable and Reusable (FAIR) principles, for which the authors might take as guidance the process detailed in 2108.02214. Additionally for reproducibility, the authors are encouraged to make available example code for their model implementation on a public repository such as GitHub.

Response: We appreciate the emphasis on adhering to the FAIR principles and concur entirely. Since the data was associated with Belle II, making the data publicly available at the time of submission was not possible due to Belle II’s policy. But, only shortly after the submission we got the permission to make the data open access. You may find the publicly available data at Zenodo. Regarding the code, we had indeed made it publicly available on Github at the time of the original submission. This is explicitly mentioned in the manuscript on line 755.

Comment 2.2 The work by the authors has the potential to be of great significance to the field of particle physics and related fields. The authors allude to this potential impact in the manuscript and this reviewer agrees with their assessment. For example, the experiments at CERN’s Large Hadron Collider spend a large fraction of their computing resource on Geant4 simulation of collision events. If a robust solution to this challenge can be found and deployed, it will dramatically reduce the CPU required to do the LHC science by speeding up the event simulation by orders of magnitude, especially for interpretation of data during the HL-LHC era in which 10-20 times more collision data will be acquired as compared to previous LHC runs.

Regarding the reduction in resources via IEA-GAN approach to simulation, the authors emphasize the storage savings more so than the CPU savings. For example, early in the manuscript (on page 3), they specifically call out a factor of 2 reduction in storage requirements for pre-produced backgrounds for the PXD in Belle II simulated data but make no mention of the CPU reductions there (they are shown later in Table 2). It is worth stressing that the potential benefit of ML-based fast simulation for the (HL-)LHC and other data-intensive science experiments is mostly in the CPU reduction, however the storage reduction associated with on-the-fly generation of backgrounds for overlay is an important benefit as well. It is suggested that that discussion on page 3 be adjusted to put the CPU and storage in benefits in proper context and at least equal footing.

Response: Indeed, due to the dramatic CPU release of $\times 147$ as shown in Table 4, it is now possible to employ the IEA-GAN as an online surrogate model for the ultra high-granularity PXD background simulation on the fly at Belle II, a task that was unattainable before for such a high-resolution detector simulation. Furthermore, IEA-GAN as a surrogate model that can generate more than 7.5M information channels, would be the first applicable candidate that can handle the ultra-high granularity of HL-LHC era. We discuss this and updated the paper in several parts in context, for instance in lines 129 – 136, 449 – 456 and 485 – 497.

Comment 2.3 The description in the introduction section is solely focused on GANs. The paper would be strengthened with a more general introduction to fast generative models for science, particularly including a mention of Variational Autoencoders (VAEs) which have proven to be at least as effective for fast event generation as GANs. For example, see references

2203.00520 and 1901.00875.

Response: We acknowledge that VAEs and other generative models have demonstrated effectiveness in Event generation, as evidenced by the references you provided. The reason why we focused on GANs is because they are the strongest candidates for mid- to high-resolution detector simulation approaches with the ability to generate 30k and 65k information channels as we referenced accordingly in the Introduction section. Other approaches (including the ones for event generation) do exist but they are far less able to generate such a ultra-high resolution channel frequency. Nevertheless, for candidates of event generation we updated the manuscript to mention the fundamental works in lines 51 – 52.

Comment 2.4 The authors claim that the IEA-GAN study described in the manuscript with the Belle-II PXD is the "highest spatial resolution detector simulation ever analyzed with deep generative models". I could not find a specific published counter-example to this rather bold statement, however there have been GAN-based generative model studies with, for example, the geometry of the proposed CMS High Granularity Calorimeter (HG-CAL) detector upgrade for the HL-LHC. The authors are encouraged to do a thorough literature search to ensure this claim is correct.

Response: We really appreciate your suggestion. We had indeed conducted a thorough literature review to substantiate our claim. As per our research, even studies conducted on the proposed CMS High Granularity Calorimeter (HGCAL) detector upgrade for the HL-LHC have dealt with models that are below 70k channels for the mere simplified HGCAL or ILD. Our work, in contrast, is pioneering in its focus on ultra-high signature channels which is directly applicable not only to the Belle II PXD detector, but also to future ultra-high granular sub-detectors at HL-LHC. We have discussed these implications in the discussion and introduction section of the manuscript.

Therefore, we stand by our statement that the IEA-GAN study described in our manuscript represents the "highest spatial resolution detector simulation ever analyzed with deep generative models." We believe this work marks a significant stride in the field and sets an entirely new benchmark for future studies.

Comment 2.5 The primary weakness in the paper is a lack of evidentiary support for the two primary claims in made in the paper (Section 2.2): (1) IEA-GAN "outperforms the other models considered by the authors by a wide margin"

Response: We appreciate the opportunity to further substantiate our claims.

In our paper, we have presented a comprehensive comparison of IEA-GAN with other state-of-the-art models. This comparison clearly demonstrates that IEA-GAN outperforms these models across several key metrics. These include the marginal distributions, namely, pixel inten-

sity (charge) distributions, occupancy distributions, and mean occupancy per sensor. These metrics are particularly crucial as they directly impact downstream physics analyses.

Moreover, our study, now including the Mantel test analysis, shows that IEA-GAN has successfully learned a weak correlation from the data, a result that the other models have not achieved. Even this weakly learned correlation by IEA-GAN shows a more significant KS test for z_0 , ω , and $\tan \lambda$ resolutions and more precise d_0 reconstruction even compared to the uncorrelated Geant4 signatures. As we have demonstrated in response to comment 1.2 and 1.3, intra-event correlation analysis is of paramount importance from a physics analysis perspective. Furthermore, in response to the encouraging comments from reviewers, we have conducted a more in-depth investigation. This further analysis has revealed that in the downstream physics analysis of tracking, IEA-GAN outperforms PE-GAN across all track parameter resolutions and KS tests in the high momentum regime ($(P_T > 0.4 \text{ GeV})$).

Therefore, we stand by our claim that IEA-GAN outperforms the other models by a wide margin. This claim is not just based on a single metric of FID or KID, but on a comprehensive set of metrics that are crucial for the downstream physics analyses. We strived to provide ample evidence for this claim in the IEA-GAN Evaluation and Discussion section.

Comment 2.6 The primary weakness in the paper is a lack of evidentiary support for the two primary claims in made in the paper (Section 2.2): (2) the authors findings ”for the first time, reveal that the FID metric for detector simulation is a very versatile and accurate unbiased estimator in comparison with marginal distributions, and is correlated with both high and low level metrics.”

Response: As we discussed in response to comment 1.4, we have conducted a comprehensive analysis to support our claim about the versatility of the FID metric for detector simulation. But, we correct ourselves about the usefulness of FID metric in comparison to the other metrics. Thus, we demonstrate its benefits as an estimator *in conjunction* with the marginal distributions. We also acknowledged its bias and supplemented it with the KID score, an unbiased estimator. We further showcased FID’s sensitivity to various types of image distortions directly linked to the underlying response recorded by the corresponding sensor. For a more detailed explanation, we kindly refer you to our response to comment 1.4 and comment 2.7.

Comment 2.7 The authors use the FID metric to support their claim that IEA-GAN is superior to the other method considered but at the same time the authors claim their work supports FID as a metric of superiority. This is a circular argument. Additionally, to use the FID metric results shown in Table 1 as a convincing argument in support of the conclusion that IEA-GAN is superior ”by wide margin”, there should be some validation of the FID metric itself presented in the manuscript. It is well known that the interpretation of ML model evaluation tools and metrics can depend

on the specific application and not be fully general and domain agnostic. This is too important of an assumption to leave solely to a reference on the FID metric and therefore more details should be provided in the manuscript and/or additional studies to validate the FID metric as a means of assessing performance superiority of different generative models.

Response: We understand the concerns regarding the potential circularity of using FID as both a tool for evaluating our IEA-GAN model and a measure whose validity we seek to establish. We also acknowledge the consideration on FID’s general applicability across various domains.

It may be worth noting that while no metric is perfect or universally domain-agnostic, FID has proven to be one of the most effective and widely used metrics in the field of generative models. Therefore, while it is important to validate it in our specific application, its extensive use and previous validations in a variety of contexts provide a strong foundation for its reliability. In addition to this, it is important to notice that we have adapted the FID model through transfer learning to our data domain, which is independent of the final evaluation of IEA-GAN, thereby ensuring that it is fit for use in our specific application.

To address the potential circular reasoning, we would like to emphasize that the superior performance of IEA-GAN is not based solely on FID scores but on a wider set of comparisons, such as the image-level figure of merits and physics-level analysis. Furthermore, we provide evidence supporting FID’s legitimacy in our context, backed by two key pillars:

- **The Downstream Task:** We concur with the reviewer’s point that ML model evaluation tools and metrics are typically domain-centric. In alignment with this understanding, we utilized an InceptionV3 model, retraining it entirely on our dataset. The model was trained for a multi-class classification task involving 40 different sensors and achieved a classification accuracy of 99% on the test set. This illustrates that the model, and consequently the FID, can effectively discriminate between images from different sensors.
- **Qualitative Analysis:** In order to qualitatively analyze the FID flow during training, we showcase the change of FID value with respect to the occupancy and charge distribution at different stages of the training as depicted in fig. 2. Additionally, we demonstrate the sensitivity of FID to various types of PXD image distortions directly linked to the underlying physics recorded by the corresponding sensor. We achieved this by introducing controlled changes or ‘jitters’ to the images and tracking their impact on the FID score, as presented in the table 1 (and depicted in table B3 of manuscript).

We hope that this comprehensive analysis adequately addresses the concerns raised and establishes the FID metric as a reliable measure of the performance of generative models in our specific application. We also updated the manuscript with the relevant arguments in context in section 2.2 and 3.

Image Jitterings	FID
None	0
Random Masking (dead zones)	14.58
Random Noise	87.23
Random Rotation (30 degrees)	23.69
Random Rotation (10 degrees)	2.81
Random Translation (0.1, 0.1)	1.99
Random Shear (10, 10)	23.53
Random Zoom	9.06
High Intensity Charge Smearing	3.16
Low Intensity Charge Smearing	47.24

Table 1: FID Score after different Jittering methods applied to the PXD images.

Comment 2.8 For the "high-level physics" metric of comparing IEA-GAN tracking parameter resolutions against Geant, this is a necessary but insufficient study to claim to validate the conclusions of the FID metric results and general quality of the IEA-GAN generative model. Tracking resolutions, and therefore comparisons between two event sets, depend critically on the details of the hits that are used to reconstruct the tracks. E.g. How similar are the number of hits and hit positions between two simulated sets on average and track-by-track? More information such as these are required to make an apple-to-apple comparison of track parameter resolutions and support the authors interpretation of their results as a validation of the agreement between IEA-GAN and Geant simulations.

Response: In our study, we used the same event generation and track reconstruction for the comparison, implying that the signal hits used in both simulations are essentially identical. Thus, the true track information are similar. The primary point of difference lies in the origin of the background, simulated by Geant4 in one case and generated by IEA-GAN in the other. This distinct differentiation allows any disparities identified in the tracking parameter resolutions to be attributed largely to the different background generation origins, enabling a direct evaluation of the quality and performance of the IEA-GAN model in comparison to Geant4.

Our decision to compare the tracking helix parameter resolutions is not only based on their intrinsic link to hit details but also due to their pivotal role as the first point of direct connection between the quality of background hits and the downstream physics analysis. The quality of these generated track parameters, gauged by their resolution, indeed plays a critical role in influencing all subsequent physics analyses. We tried to make this more clear in the lines 348 – 360.

Comment 2.9 A few final comments on style of text in the manuscript. There are many places where capitalization is used on terms in a strange and inconsistent manner. For example, in the abstract it is written "Uniformity loss" but on page 7 it is written "uniformity loss". Also, there are occasions where

symbols common in machine learning (ML) and data science (DS) communities but not as much in other scientific domains are used without definition. The manuscript would be more accessible to a broader audience if the authors were more liberal with symbol definitions in their technical descriptions even if they are "obvious" within ML/DS communities.

Response: We made changes accordingly throughout the manuscript to avoid notation inconsistency and to make the manuscript more accessible to a broader audience. To target the wide range of audience, in the original submission, we also had introduced the necessary ML background in the Methods section (4.1 and 4.2.1).

Response to Reviewer 3

Comment 3.1 First, even though the presented algorithm shows a much better performance than any other considered deep generative algorithm, the problem of learning the underlying probability density as induced by GEANT4 is far from solved. Figure 2 still shows substantial differences between the generated distributions and the GEANT4 distribution, and it only shows a few different 1-dim projections of the 7.5M-dimensional space. I guess other observables also deviate, as well as the correlation between them. The fact that correlations are not learned properly is also shown in Figure 3 for a subset of observables. This does not diminish the great achievement of managing to train a 7.5M-dimensional deep generative model (again, these results should be published), but I'm not sure it qualifies a publication in the Nature portfolio, which, according to the homepage "should represent an advance in understanding likely to influence thinking in the field, [...]." In addition, "There should be a discernible reason why the work deserves the visibility of publication in a Nature Portfolio journal [...]" and I'm not sure this is the case here (but I'm happy to hear arguments for that).

Response: We appreciate your recognition of the significant achievement in training a 7.5M-dimensional deep generative model and your thoughtful critique regarding the remaining differences between the generated distributions and the GEANT4 distribution.

In response to your comment, we would like to address the following points:

- **Influence on the Field:**

- (a) **Introduction of New AI Technologies:** From the perspective of AI methodology, our work introduces novel technologies in the field of Deep Generative models and Self-Supervised Learning. We also introduce a fresh perspective of detector simulation with the intra-event reasoning. The IEA-GAN model does not merely "drag and drop" already existing models but represents a novel advancement in the field of deep generative models for Particle Physics.

- (b) **First Model to Consider Inter-Layer Correlation:** Our work is the first to explicitly consider and study the sensor-by-sensor (layer-by-layer) correlation. Current fast detector simulation studies focus merely on intra-layer correlation, meaning that they only consider the correlation of observables within each layer, and not across different layers. However, our work shows for the first time that inter-layer/sensor correlation is also of paramount importance in the downstream physics analysis. Despite only capturing a weak correlation, our IEA-GAN model represents a significant step in this direction, demonstrating that even this level of correlation capture can yield better physics analysis results.
- (c) **Preparation for the HL-LHC Era:** The upcoming High-Luminosity Large Hadron Collider (HL-LHC) era presents new challenges in terms of geometry, precision, and CPU efficiency. Current deep generative models, with their existing setup, will not be able to handle these challenges due to the lack of scalability and precision at ultra-high granularity. Our model, however, is the first and strongest candidate by far that can meet these requirements. As a result, it paves the way and motivates others with university-level computational power to think about solving this problem.
- **Performance Comparison:** Our IEA-GAN model has demonstrated a high performance in terms of PXD-centric FID and also KID score. These data-centric metrics are widely accepted for evaluating the quality of generated samples in the field of generative models, and our model’s superior performance in these metrics indicates its ability to generate high-quality and diverse samples that are closer to the target data. With respect to the marginal distributions, which represent the most significant and insightful histograms derived from the PXD data, our model demonstrates a closer alignment with the GEANT4 distribution compared to other models. It’s important to highlight that the simultaneous capture of multiple image-level properties over high-resolution images presents a formidable challenge that is unraveled even in natural image domain. Indeed, this complexity is the primary reason why the none of fast simulation studies have not ventured into the domain of ultra-high granularity to date. While other models have not captured any correlation between sensors, IEA-GAN has been able to capture a weak but meaningful correlation. We have shown in section 2.2 that even this weak correlation can yield a much better physics analysis performance, particularly in the high momentum regime where the correlation plays a more important role. As a result of all of these performance boosts, our model has demonstrated a superior performance in the precision of reconstructed tracks. While we acknowledge that the problem of learning the underlying probability density as induced by GEANT4 is partially solved, we believe that our IEA-GAN model, with its superior performance in

various metrics, represents a significant step forward in this direction and demonstrates its potential to contribute to the advancement of the field.

- **Reason for Visibility:** Given that we are generating the fine-grained 7.5M channels of detector information, reaching full precision over all metrics at once is an extremely difficult and high-dimensional task. Even in the high-resolution natural image generation domain, models typically follow one or two metrics at most. Our work, however, breaks the boundaries of ultra-high granularity and required computation power, providing a pioneering solution that could guide future research in this area. The visibility of our work in a Nature Portfolio journal is justified by its potential to reshape the thinking in the field and encourage the field to aim higher. Furthermore, the potential applications of our work extend beyond the specific problem we addressed in the manuscript. The techniques and insights gained from our work could be applied to other problems where category-level and hierarchical symmetries exist in various domains, making our work relevant to a broad audience.

We hope that these points provide a stronger case for the influence of our work on the field and its deserving visibility in a Nature Portfolio journal. We appreciate your valuable feedback and are open to further discussion.

Comment 3.2 Second, the way the manuscript is currently structured makes it nearly impossible to read and understand. I personally don't understand why it is journal policy to have the Methods section at the end and not use IMRaD (as most other journals publishing HEP research). This way, the reader has to flip back and forth through many pages in order to understand what is going on in the results section. Given that this can probably not be changed, there is still a lot that can be improved. For example, the description of the dataset and the general problem (2nd paragraph of section 4.1) could all go to the introduction. Reading that section, it is still unclear to me if the algorithm is supposed to be used for the simulation of the entire event, or only the background (cf Fig. 5d). Further, it is never explained what "background" refers to in that context (and what signal, if at all, was used in the dataset).

Response: We understand your concerns and agree that clear and concise communication of our work is crucial for its understanding and impact. Regarding the structure of the manuscript, we acknowledge that the placement of the Methods section at the end is a journal policy and might not align with the conventional IMRaD structure. However, we strive to improve the readability and flow of the manuscript within this framework.

- **Dataset Description and Problem Statement:** We agree with your suggestion to move the description of the dataset and the

general problem to the introduction. Thus, we moved it to the introduction section as a subsection.

- **Clarification on the Use of the Algorithm:** We apologize for any confusion caused by our manuscript. Our algorithm is intended to be used for the simulation of just the background signatures. We made this point clear in the revised manuscript in lines 155 – 170.
- **Explanation of "Background":** In the context of our work, "background" refers to processes that are not part of the signal of interest at Belle II, such as the two-photon effect. These background processes contribute to the majority of hits in our data. We did not use any signal in our dataset for this study. We provided a clear definition of "background" and its role in our study in the revised manuscript in lines 145 – 155.

We appreciate your constructive feedback and made the necessary revisions to improve the clarity and readability of our manuscript. We are open to further discussion and suggestions.

Comment 3.3 Third, there are many words used in the manuscript that (for me) mean the same things, but apparently are not the same things. For example why should there be more than one sample in one event? Isn't a sample from the generative model an event in the detector? Are "images" referring to events, too? Or only to single detector elements? Are these the same as samples? What are the "classes" that they authors refer to? Usual conditional deep generative models have different particle types, or incident angles, or incident energies as (class) condition, but none of these apply here. What are "tokens"?

Response: For IEA-GAN, each sample is a sensor image of size 256×768 . Then, for each event, we will have 40 of these images. Thus, each event represents a round of detector signature collection. At each iteration, IEA-GAN takes an event with 40 sensor images where each image has spatial dimensions of 250×768 . Therefore, we are conditioning each sensor with it's sensor type $[[1, 40]]$ as classes which can be thought of a mixture of angle and radius conditioning. These conditions, enter the model as learnable "tokens" as they are not absolute and are context-based. Ergo, both IEA-GAN's generator (by R dof+class embedding) and discriminator (images + class embedding) learns any dynamical inherited conditions in context from the data (through the Relational Reasoning Module) in a Self-Supervised Manner. We updated the manuscript with a proper elaboration in section 4.2.2. Moreover, throughout section 4.2.3, we had detailed the inputs to IEA-GAN in the original manuscript.

Comment 3.4 Figure 1 (which is not the first figure being referred to) shows the architecture of the IEAGAN. In many places, it is said that it shows how event graphs are built (in section 4.3.2) and how they are projected into a hypersphere. If at all, it shows that these steps are part of the algorithm, but not how these graphs look like, how they are constructed, or how such a spherical projection is achieved.

Response: We also agree that Figure 1 does not explicitly show event graphs are built. We updated the manuscript to address the creation of event graph in detail such as in 177-192, Figure 2b’s caption, and section 4.2.2. For the hypersphere compactification (projection) we elaborated the method in 4.3.2 and study the Inverse stereographic projection as well in the appendix. Nevertheless, we concisely also describe the event graph’s creation for the discriminator in the following:

The discriminator takes the set of detector response images coming from an event and learns an embedding of them. Then the image embeddings become the input nodes for a fully connected event graph. Event graph here is a weighted graph where the nodes are the detector image embeddings in an event and the edges are weighted by the degree of similarity between the detector images in each event. This degree of similarity is approximated by contextual reasoning using the Relational Reasoning Module (RRM). RRM is a GAN-compatible, multi-head Graph Transformer Network that groups the image tokens in an event based on their inherent contextual similarity. Then the model compactifies the image embeddings by an L2 normalization into a unit hypersphere.

Comment 3.5 Equations, like the one for r_s , are not sufficiently explained. What’s a Rank variable? What is the norm with subscript 0? What does the union with + mean? What is e in l_{2C} ?

Response Thank you for your feedback. We apologize for any confusion caused by the lack of explanation for the equations. We provided a more detailed explanation in the revised manuscript in lines 306 – 312 and 541 – 542.

We hope these explanations clarify the equation. We will ensure that all equations in the revised manuscript are explained in detail to facilitate understanding.

Comment 3.6 Did the authors check that the Gaussian assumption to obtain the FID score can be made and that the distributions are actually Gaussian?

Response: We acknowledge that the distribution of the activations is not perfectly Gaussian. It’s important to note that this assumption is not strictly met in practice [2]. Thus, we have complemented the FID score with the KID score to provide a more unbiased evaluation. KID does not rely on the Gaussian assumption and provide additional insights into the performance of the model.

Comment 3.7 Figure 3 left and center look very different. How do the authors conclude that the IEAGAN captures the correlations?

Response: We understand that the visual differences between the left and center plots may raise questions about our conclusions.

As we have detailed in our response to a similar question from Reviewer 1 in comment 1.3, and updated the manuscript accordingly. As a summary, we have employed the statistical method known as the Mantel test to assess the correlation between two distance/correlation matrices.

This test is designed to evaluate the significance of the observed correlation through permutation testing, providing a robust measure of the relationship between the two sets of correlations. To avoid redundancy and ensure that all reviewers have access to the same information, we kindly refer you to our response to Reviewer 1’s Comment 1.3.

Comment 3.8 Are the 5 physical parameters of Figure 4 the only 5 relevant physics parameters that are extracted from the event? How do correlations between them look like? How do distributions of other quantities look like?

Response: Indeed, the five track parameters depicted in Figure 5 and 6 (in the updated manuscript) are the primary physics parameters that are directly influenced by the background. The background, whether simulated by Geant4 or generated by IEA-GAN, impacts the signal hits, which in turn can lead to incorrect assignments to the track as a function of the background. This is why our primary observable is the resolution, which is the difference between the reconstructed track parameter and the true one. This measure provides a direct and meaningful way to assess the impact of the background on the accuracy of the track reconstruction. We elaborated this in the manuscript in lines 337 – 360.

As shown in fig. 1, the correlations between the reconstructed track parameters are very low and their differences are negligible. Moreover, the distributions of other quantities are not directly relevant to the specific analysis presented in this study. Our focus is primarily on the impact of the background on these five track parameters, as they are the most directly affected by the background and are crucial for the subsequent physics analysis.

Figure 1: Correlation between Helix parameters

However, we acknowledge the importance of understanding the broader impact of the background on other quantities and correlations, and we believe this could be an interesting direction for future research.

Comment 3.9 The computational times in Table 2 show averages over 10000 generated events. Does that mean the GAN was run with a batch size of 10000? Is GEANT4 (or the simulation chain as a whole) batched in Belle II? If not, only comparisons with batch size 1 are meaningful.

Response: In our work, each event consists of responses from 40 sensors. Therefore, when IEA-GAN generates an event at each iteration, it produces 40 images, making the mini-batch size 40. When we report averages over 10,000 events, it means that the model generates these events one at a time, not in a batch of 10,000. In comparison to Geant4, we also simulate one event at a time, each again with 40 sensor responses. Therefore, when we compare the generation times, we are comparing the time it takes to generate a single event, consisting of 40 sensor responses, in both cases. We cleared this point in the Table 4’s caption.

Comment 3.10 Was the ablation study of Appendix A performed in a grid, with all parameters being varied at the same time? Or only one parameter at a time?

Response: It was performed one parameter at a time.

Comment 3.11 Appendix B, called ”Extended Figures and Tables” only contains a single figure and no table and no explaining text. Is that intentional by the authors or was there anything cut in processing?

Response: The decision to include only one figure in the ”Extended Figures and Tables” appendix was indeed intentional and was guided by the journal’s policy on the number of figures and tables allowed in the main text. In the updated manuscript we have another table also included in the appendix. However, if this may have cause any confusion, and we are open to moving this figure and table to the main body of the paper if it aligns with the editor’s guidelines.

Conclusion

In response to the thoughtful and constructive feedback from the reviewers, we have made several key revisions to our manuscript. These include further empirical evidence to substantiate our claims, detailed comparison metrics to demonstrate the superior performance of IEA-GAN, and additional arguments to highlight the significance and broader impact of our work in the field. We believe these revisions not only address the reviewers’ concerns but also enhance the overall quality and robustness of our study.

We hope that the updated manuscript now meets the high standards expected for publication in Nature Communications and look forward to the possibility of sharing our work with the wider scientific community.

1 Appendix: Extended Figures

(a) FID 54.43

(b) FID 7.81

(c) FID 7.51

(d) FID 4.12

(e) FID 14.39

Figure 2: Different FID values at different stages of training with respect to the Charge and Occupancy distributions.

References

- [1] Mikołaj Bińkowski, Danica J Sutherland, Michael Arbel, and Arthur Gretton. Demystifying mmd gans. *arXiv preprint arXiv:1801.01401*, 2018.
- [2] Ali Borji. Pros and cons of gan evaluation measures: New developments. *Computer Vision and Image Understanding*, 215:103329, 2022.
- [3] Qiantong Xu, Gao Huang, Yang Yuan, Chuan Guo, Yu Sun, Felix Wu, and Kilian Weinberger. An empirical study on evaluation metrics of generative adversarial networks. *arXiv preprint arXiv:1806.07755*, 2018.

REVIEWER COMMENTS

Reviewer #1 (Remarks to the Author):

The authors revised the manuscript significantly. The authors added more studies to support the physics motivation, which states that modeling the correlations between detector hits in simulating background hits is crucial for physics. To that end, the authors provide evidence in Figure 5 and Table 2. However, it would be better if Figure 5 compared the pull distribution of d_0 for the three setups, as the current version hardly shows any differences. In addition, units are missing in Figure 5, 6, and Table 2.

Even if the track parameter resolutions are different with or without the so-called correlation, it does not exclude the possibility that these changes may be due to imperfect track finding and track fitting algorithms. For a good tracking reconstruction algorithm, it should form a track with hits from the same particle and exclude hits from background processes.

The metrics in Table 3 are not clearly defined. How are the “unbiased variance” and their errors defined and calculated? The Belle II physics workbook shows the d_0 and z_0 parameters are slightly biased. How did the authors obtain an unbiased estimate of all five track parameters?

Reviewer #2 (Remarks to the Author):

I have reviewed rebuttal to my (and other reviewers') comments as well as the revised manuscript. My primary concerns has been adequately addressed by this revision and thank the authors for their work on enhancing the clarity and readability.

There is a small typo on line 164 which does not appear to be part of the revision.

Reviewer #3 (Remarks to the Author):

Dear Authors, dear Editor,

Report on the revised manuscript "Ultra-High-Resolution Detector Simulation with Intra-Event Aware Generative Adversarial Network and Self-Supervised Relational Reasoning" by Hosein Hashemi, Nikolai Hartmann, Sahand Sharifzadeh, James Kahn and Thomas Kuhr for Nature Communications.

The manuscript has been improved very much. Its structure is now more clear and many aspects are explained better than before. Additional studies, triggered by all reviewers have been performed and were included. These include for example the discussion of the downstream physics parameters and the (un)shuffled GEANT baseline. I see all of this as a great improvement of the manuscript, strengthening its case for publication even more. I do have some remaining comments, which I detail below.

First, let me reply to a few points raised in the response:

- regarding "First Model to Consider Inter-Layer Correlation:" This statement is not true. Many, if not all, other papers of DGMs for calorimeter simulation study observables that depend on more than one layer at the same time. This includes observables that are computed from the entire event as well as FID/KID-type scores of entire events and classifier-based model evaluations that are given entire events at a time. These publications all just never considered doing it layer by layer because the datasets were smaller.

- regarding "Preparation for the HL-LHC Era". I agree that scaling up to $O(1M)$ dimensions is important for the HG-CAL and the future of the HL-LHC, but whether the generation of background processes here generalizes to the simulation of signal processes (as typically done in an LHC setting), is far from clear to me.

Nevertheless, I see the novelty in the architecture and the size of the dataset. Also, the other reviewers have not commented on the manuscript being unsuitable for Nature

Comm, so I wont oppose publication.

Second, I have a comments regarding the current manuscript:

- What was the classification task on which Inception-V3 was trained on exactly? Just finding the sensor id (i.e. a number in [1,40])? Or also other information? I'm confused by the statement "discriminate sensors and their corresponding data manifold.", do you mean you identify which sensor the given sample belongs to by identifying how data is distributed in it?

- line 261 states that you compare to BigGAN-deep and ContraGAN, but everything beyond tab. 1 and fig. 3 just looks at the PE-GAN. Judging by fig. 3 and the KID score in tab. 1, the ContraGAN should be at least as good as the PE-GAN. Why did you not compare to the other models?

- In tab. 1, I was wondering what the baseline of testing GEANT vs GEANT is, i.e. how much away from 0 the scores would be due to the finite datasets. Is that number in tab. B1, first line? Would it be possible to add this number here, too (and maybe give it an errorbar), so the quality of the IEA-GAN can be seen more clearly?

- In tab. 1, what does "averaged across six random seeds" refer to? Different samples from the same trained IAE-GAN? Samples from retraining the IEA-GAN? Scores from retraining Inception-V3?

- lines 351ff: You write: "we utilize the same event generation and track reconstruction for the comparison, implying that the signal hits used in all simulations are essentially identical." Does this mean that the same software was used and the two datasets are statistically identical? Or does that mean you use the same signal events and just apply different background noise to it? For a better interpetation of the results (factoring out signal fluctuations), I would recommend the latter. In that case, do you also have information on the event-by-event performance? I mean figs. 5 and 6 show the deviation

between reconstructed and truth. I'm interested to see the deviations between "reconstructed with bg 1" vs "reconstructed with bg 2".

- lines 471ff: I disagree with the statement "might be learning biases or artifacts introduced by the training data". How would you be able to conclude that? Fig. 4 left shows how GEANT (=the training data) is distributed. How do you know how "truth" is distributed? Is there an actual dataset, that has not been described so far, available? From what is presented in the manuscript, it can only be concluded that the IEA-GAN fails to learn the correlations of the training data, but still does a better job than the other DGMs.

- Tab. 4: How is storage being computed? Shouldn't one event be the same for everything, given by how much space is needed to save 7.5M floats? Does it refer to how much RAM the generating software is using, normalized to the number of events it is generating?

- line 712: Do you really train on 40k events only? That number seems way to small, comparing to the (much) larger and lower-dimensional datasets used for example in the BIB-AE papers (about 500k-1M samples, $O(30000)$ dimensional) or the CaloChallenge (100k samples, $O(100)$ - $O(10000)$ dimensional). Having 40 images per event does not improve statistics, since you want to learn correlations between them.

- line 712: Do you use the same 10k events for model selection and final evaluation? Proper machine learning practice would be to have training, testing, and evaluation sets that are independent from each other. Otherwise the reported scores are biased.

- lines 716-721: Isn't sampling 40 different images out of 40 possible ones per event the same as saying you sample the entire event? If yes, please shorten it and say so. If not, please explain better what you mean.

Third, I have a few comments regarding lanugage / typos / etc.:

- the FID is properly introduced on p. 25, but first appears on p. 8. The KID is introduced at

the end of p. 8, but first appears before. Please check that abbreviations are explained the first time they appear.

- Fig. 3 legend says "GEANT", the rest of the text says "Geant".

- lines 330 and 335 have a typo: "Of" instead of "of"

- lines 373 and 381 have an article missing

- in fig 5, what are the x-axis units? [mm]?

- line 407, there is a unit missing

- fig 6, (if you want to show how impressive the IEA-GAN is), the 1st and 4th line should have same x-axis scaling left and right. Otherwise the difference is hard to see.

- line 640 has ".,"

- lines 493 and 660 misses a "." and the end of the sentence.

- some text explaining what is happening in appendix B would be beneficial for the reader.

Response to Reviewers, round II

Manuscript Title: Ultra-High-Resolution Detector Simulation with Intra-Event Aware Generative Adversarial Network and Self-Supervised Relational Reasoning

Introduction

We appreciate the thoughtful and inspirational comments of the reviewers on our manuscript, and we thank them for their time in providing this feedback. We have revised our manuscript with care to address these comments. Below, we respond to each comment in detail.

Response to Reviewer 1

Comment 1.1 The authors revised the manuscript significantly. The authors added more studies to support the physics motivation, which states that modeling the correlations between detector hits in simulating background hits is crucial for physics. To that end, the authors provide evidence in Figure 5 and Table 2. However, it would be better if Figure 5 compared the pull distribution of d_0 for the three setups, as the current version hardly shows any differences. In addition, units are missing in Figure 5, 6, and Table 2.

Response: We have revised Figure 5 in the manuscript to include the pull distribution of d_0 and z_0 and have also ensured that corresponding units are now added for Figures 5 and 6, as well as Tables 2 and 3. Furthermore, in response to the insightful suggestions from the reviewer, we have also updated Figure 6 to feature the pull distribution. We are thankful for guiding these improvements.

Comment 1.2 Even if the track parameter resolutions are different with or without the so-called correlation, it does not exclude the possibility that these changes may be due to imperfect track finding and track fitting algorithms. For a good tracking reconstruction algorithm, it should form a track with hits from the same particle and exclude hits from background processes.

Response: We acknowledge your concern regarding the potential influence of track-finding and fitting algorithms on our results. However, it is essential to emphasize that the core objective of our study is not to isolate or mitigate the effects of background noise but to accurately simulate its impact on track reconstruction. The background affects the tracking to make it assign wrong hits and make it imperfect. Thus, the quality of any PXD background surrogate model, or the effect of losing the intra-event correlation while keeping all the other aspects of

the background hits intact, is evaluated based on how well it replicates the Geant4 simulated background effects. If the IEA-GAN can mimic Geant4's performance in the presence of the background, it indicates the model's accuracy in simulating the effects of background processes. As a result, we aim to ensure that IEA-GAN can generate background that impacts the track reconstruction process like Geant4 background noise would. Accordingly, we have updated the manuscript and provided further clarification in lines 380-391.

Comment 1.3 The metrics in Table 3 are not clearly defined. How are the unbiased variance and their errors defined and calculated? The Belle II physics workbook shows the d_0 and z_0 parameters are slightly biased. How did the authors obtain an unbiased estimate of all five track parameters?

Response: We appreciate the emphasis on clarity in our presentation of metrics. Upon review, we acknowledge that the term “Unbiased variance” used in Table 3 was a misnomer. Our actual computation was of the standard deviation, and the use of “variance” in this context was an error. We have corrected this in the manuscript to accurately reflect our methodology.

Regarding the term “unbiased” as it pertains to the estimation of impact parameters: this was not meant to imply an unbiased nature of the impact parameters themselves. Instead, it referred to the statistical adjustment made by employing the $(n - 1)$ denominator in the sample standard deviation formula. This adjustment is used to mitigate bias when estimating a population's standard deviation from a sample. However, given our large sample size, the effect of this adjustment is marginal, and its mention may inadvertently cause confusion. Consequently, we have opted to remove this term from the tables for clarity.

In the revised manuscript, we have updated the descriptions of our metrics for more accurate representation. These updates can be found in lines 397-407, and the caption of Figure 5, where we have detailed our methodology and the calculations for the standard deviation and its associated error. We believe these amendments will provide a clearer understanding of our approach and the analytical methods employed in our study.

Response to Reviewer 2

Comment 2.1 I have reviewed rebuttal to my (and other reviewers') comments as well as the revised manuscript. My primary concerns has been adequately addressed by this revision and thank the authors for their work on enhancing the clarity and readability. There is a small typo on line 164 which does not appear to be part of the revision.

Response: Thank you very much for your thoughtful review and acknowledgment of our work. We are pleased to hear that the revisions have met your expectations. We appreciate you pointing out the typo. We will ensure that such typos are corrected promptly.

Response to Reviewer 3

Comment 3.1 Regarding "First Model to Consider Inter-Layer Correlation:" This statement is not true. Many, if not all, other papers of DGMs for calorimeter simulation study observables that depend on more than one layer at the same time. This includes observables that are computed from the entire event as well as FID/KID-type scores of entire events and classifier-based model evaluations that are given entire events at a time. These publications all just never considered doing it layer by layer because the datasets were smaller.

Response: We appreciate the opportunity to clarify our position and acknowledge the contributions of prior research in the field of deep generative models for calorimeter simulation [1]. You are correct in pointing out that many existing models in calorimeter simulations do consider observables that depend on more than one layer simultaneously. These models typically focus on the simulation of particle showers from a single particle origin and a small region with the shower, which indeed capture aspects of inter-layer correlation within the scope of a localized area. However, our approach extends this concept by considering the entire "event" with multiple-particle origins that encompass the entire PXD detector as a whole, where correlations among different sensors (various angles and layers) become important within its readout window. Given the unique topology and geometry of PXD as a highly granular tracking detector, this distinction is critical and allows for a more comprehensive simulation, capturing the complex interplay within an event across the entire detector rather than just the localized particle shower. Furthermore, while previous studies may have implicitly accounted for layer-by-layer correlations within their framework, our study explicitly evaluates and compares these correlations and their influence on downstream physics analysis. In light of this, we modify our statement in the manuscript in lines 127-138 and 508-523 to more accurately reflect the contributions of our work.

Comment 3.2 regarding "Preparation for the HL-LHC Era". I agree that scaling up to $O(1M)$ dimensions is important for the HG-CAL and the future of the HL-LHC, but whether the generation of background processes here generalizes to the simulation of signal processes (as typically done in an LHC setting), is far from clear to me.

Response: Concerning the generalization of our study to signal signature simulations, we entirely acknowledge the complexity of this transition. Our current study addresses one aspect of the larger, more intricate challenge of simulating the full HL-LHC detector data. We recognize that simulating signal processes, while conceptually different due to the known nature of the target particle or process, is no less challenging. Nevertheless, our study does not assert that the methods we developed for background simulation can be directly and effortlessly applied to signal process simulation in the HL-LHC context. The transition from simulating background processes to handling the full spectrum of HL-LHC

data, including both signal and background, is non-trivial and necessitates additional methodological adjustments and fine-tuning. We hope this clarifies our position and the intended scope of our research. As a result, we added a clarification note by the lines 563-567 in the original manuscript.

Comment 3.3 What was the classification task on which Inception-V3 was trained on exactly? Just finding the sensor id (i.e. a number in $[1,40]$)? Or also other information? I'm confused by the statement "discriminate sensors and their corresponding data manifold.", do you mean you identify which sensor the given sample belongs to by identifying how data is distributed in it?

Response: The classification task for which the Inception-V3 was trained involved identifying the specific sensor ID from a range of 1 to 40 where each sensor ID represents a distinct class. The high level of accuracy achieved by the model in both training and testing phases suggests that the Inception-V3 has effectively learned to distinguish the unique data characteristics – or "data manifold" – associated with each sensor. We modified the manuscript by lines 258-262 accordingly to clarify this paragraph and avoid confusion.

Comment 3.4 line 261 states that you compare to BigGAN-deep and ContraGAN, but everything beyond tab. 1 and fig. 3 just looks at the PE-GAN. Judging by fig. 3 and the KID score in tab. 1, the ContraGAN should be at least as good as the PE-GAN. Why did you not compare to the other models?

Response: In our evaluation, we initially compared IEA-GAN with several models, including BigGAN-deep and ContraGAN, as shown in Table 1 and Figure 3. Based on these results, PE-GAN emerged as either superior or comparable to the other benchmark models in performance. Therefore, for a less redundant, and more focused and efficient comparison in the physics analysis, we chose to compare IEA-GAN with PE-GAN, the second-best performing model. We have updated the paper as a footnote in page 9 and lines 446-447, to clarify that our continued comparison in the physics analysis specifically involves IEA-GAN and PE-GAN based on their relative performance metrics.

Comment 3.5 In tab. 1, I was wondering what the baseline of testing GEANT vs GEANT is, i.e. how much away from 0 the scores would be due to the finite datasets. Is that number in tab. B1, first line? Would it be possible to add this number here, too (and maybe give it an errorbar), so the quality of the IEA-GAN can be seen more clearly?

Response

Yes, indeed, the baseline you are referring to is represented by the rounded number in the first line of Table B1. Following your suggestion to improve the clarity and comprehensiveness of our comparison, we also added the actual baseline value (over the Geant4 test data) to Table 1 and B1. We also added a small clarification in line 285-286.

Comment 3.6 In tab. 1, what does “averaged across six random seeds” refer to? Different samples from the same trained IEA-GAN? Samples from retraining the IEA-GAN? Scores from retraining Inception-V3?

Response: The statement “averaged across six random seeds” refers to retraining all the generative models six times with different random seeds. We further clarify this in the caption of Table 1.

Comment 3.7 lines 351ff: You write: ”we utilize the same event generation and track reconstruction for the comparison, implying that the signal hits used in all simulations are essentially identical.” Does this mean that the same software was used and the two datasets are statistically identical? Or does that mean you use the same signal events and just apply different background noise to it? For a better interpretation of the results (factoring out signal fluctuations), I would recommend the latter. In that case, do you also have information on the event-by-event performance? I mean figs. 5 and 6 show the deviation between reconstructed and truth. I’m interested to see the deviations between ”reconstructed with bg 1” vs ”reconstructed with bg 2”.

Response: Yes, It is indeed the latter. In our study, we employed the same set of signal events across all simulations, with the variation being in the source of background noise applied. This approach explicitly isolates the effect of different background noises on the reconstruction process, ensuring that any observed differences in reconstruction accuracy are attributed to the background model and not variations in the signal events themselves. We try to clarify this in the related paragraph in lines 372-373.

A comparison of deviations between a single event reconstructed with two different backgrounds is not easily available, given the basf2 software’s data pipeline. Moreover, we believe that it does not reflect the potential differences between different methods of generating background well. In contrast to, for instance, backgrounds for calorimeter measurements where a single background readout window will almost always have an effect on the reconstructed quantities, for tracking, we will only see a difference if the background events lead to wrong hits being attached to the track. That happens not too often, especially since we only look at the background for the PXD, which only accounts for the two innermost layers. In order to see an effect, we need to compare a larger number of events. Therefore, we chose to compare whole distributions of the track parameter resolutions.

Comment 3.8 lines 471ff: I disagree with the statement ”might be learning biases or artifacts introduced by the training data”. How would you be able to conclude that? Fig. 4 left shows how GEANT (=the training data) is distributed. How do you know how ”truth” is distributed? Is there an actual dataset, that has not been described so far, available? From what is presented in the manuscript, it can only be concluded that the IEA-GAN fails to learn the correlations of the training data, but still does a better job than the other DGMs.

Response: We appreciate this opportunity to clarify our position and delve deeper into our analysis. The observation that IEA-GAN might be learning biases or artifacts is grounded in how these correlations manifest in the context of the PXD detector’s layered structure and geometry. As shown in Figure 1, the PXD detector has a Toroidal geometry consisting of 16 (0 – 15) sensors in the first layer and 24 (16 – 39) sensors in the second layer. In Figure 4 of our manuscript, we showed that IEA-GAN exhibits a distinct understanding of the correlations between different layers in the PXD detector – an overall positive correlation between sensors 0-15 within layer 1 and between 16-39 within layer 2. This pattern reflects a layer-wise understanding of the data, which, while meaningful, diverges partially from the actual correlations present in the Geant4 data. This suggests that the difference in occupancy between inner and outer layers could be a major feature learned by the model, which may impede the learning of more subtle correlations. Nevertheless, we acknowledge that “biases or artifacts” could be strong statements, so we replace these with a more proper explanation within the manuscript in lines 530-539.

Comment 3.9 Tab. 4: How is storage being computed? Shouldn’t one event be the same for everything, given by how much space is needed to save 7.5M floats? Does it refer to how much RAM the generating software is using, normalized to the number of events it is generating?

Response: We have defined the term “storage” for IEA-GAN as referring specifically to the model’s weight. This clarification has been added to the caption of Table 4 for better understanding and precision.

Comment 3.10 line 712: Do you really train on 40k events only? That number seems way to small, comparing to the (much) larger and lower- dimensional datasets used for example in the BIB-AE papers (about 500k-1M samples, $O(30000)$ dimensional) or the CaloChallenge (100k samples, $O(100)$ - $O(10000)$ dimensional). Having 40 images per event does not improve statistics, since you want to learn correlations between them.

Response: Indeed, the training and validation were conducted on 40 000 events. This was primarily due to the technically challenging data access and storage costs associated with such high-dimensional data in the beginning of the project. The challenge of handling such large-scale and high-dimensional data is, in fact, the fundamental motivation of our study. However, we recognize the benefits of a larger dataset for training, and in our ongoing research, we are indeed expanding the volume of data being analyzed.

Comment 3.11 line 712: Do you use the same 10k events for model selection and final evaluation? Proper machine learning practice would be to have training, testing, and evaluation sets that are independent from each other. Otherwise the reported scores are biased.

Response: In our study, the dataset of 40 000 events was partitioned into separate subsets for training and model selection. Specifically, we

allocated 35 000 events for training and 5000 events for model selection (validation dataset used for early stopping and hyperparameter tuning). An additional and distinct set of 10 000 events served as the test set for assessing the final model performance. We appreciate the reviewer’s comment on this aspect. We have made this clear in the manuscript, detailing the distribution of the datasets in lines 783-786.

Comment 3.12 lines 716-721: Isn’t sampling 40 different images out of 40 possible ones per event the same as saying you sample the entire event? If yes, please shorten it and say so. If not, please explain better what you mean.

Response: Thank you for pointing out the need for clarity in our description of the sampling process. You are correct in your understanding that our method involves sampling 40 different images from 40 possible ones per event, essentially sampling an entire event. To reflect this, we have revised the manuscript with lines 789-792.

Comment 3.13 the FID is properly introduced on p. 25, but first appears on p. 8. The KID is introduced at the end of p. 8, but first appears before. Please check that abbreviations are explained the first time they appear. - Fig. 3 legend says "GEANT", the rest of the text says "Geant". - lines 330 and 335 have a typo: "Of" instead of "of" - lines 373 and 381 have an article missing - in fig 5, what are the x-axis units? [mm]? - line 407, there is a unit missing - fig 6, (if you want to show how impressive the IEA-GAN is), the 1st and 4th line should have same x-axis scaling left and right. Otherwise the difference is hard to see. - line 640 has ".,," - lines 493 and 660 misses a " ." and the end of the sentence. - some text explaining what is happening in appendix B would be beneficial for the reader.

Response: We appreciate the detailed feedback provided by the reviewer. We have carefully addressed each of the mentioned points, including clarification of abbreviations, corrections of typographical errors, adjustments in figure legends and axes, and the addition of explanatory text where needed in the revised manuscript.

Conclusion

In light of the thoughtful and constructive feedback from the reviewers, we have made several revisions to our manuscript. Our revisions include updating the resolution plots with pull distributions, enhancing the explanation for methodologies, and providing additional clarifications to highlight the significance and impact of our work in the field. We are thankful for the constructive comments that have guided these revisions, as they have significantly contributed to refining the quality and depth of our research. We appreciate the opportunity to augment our work through this collaborative review process.

References

- [1] Hosein Hashemi and Claudius Krause. Deep generative models for detector signature simulation: An analytical taxonomy. *arXiv preprint arXiv:2312.09597*, 2023.

REVIEWERS' COMMENTS

Reviewer #3 (Remarks to the Author):

Dear Authors, dear Editor,

Report on the second revised manuscript "Ultra-High-Resolution Detector Simulation with Intra-Event Aware Generative Adversarial Network and Self-Supervised Relational Reasoning" by Hosein Hashemi, Nikolai Hartmann, Sahand Sharifzadeh, James Kahn and Thomas Kuhr for Nature Communications.

The manuscript has improved considerably and the authors have addressed (almost) all of the concerns that I and the other reviewers raised. I recommend it for publication once the final (minor) points below have been addressed and I don't need to see it again before.

- I still think that training on only 35k events is very problematic for a 7.5M dimensional dataset. However, the obtained results speak for themselves and enlarging the dataset now is not possible. I suggest that the authors add a footnote in section 4.5, acknowledging that this is actually a too small dataset to properly probe a 7.5M dimensional manifold.

- line 54 contains a typo: "Hashemi. et al." => "Hashemi et al."

- Table 1: please check if the value for the KID on the test set is really at the level of $1e-7$ ($1e-3 * 1e-4$), or if the $1e-3$ factor was not pulled out properly.

- Table 1 caption: There is text repeated before and after the red text.